# Disparity in temporal and spatial relationships between resting-state electrophysiological and fMRI signals

**Wenyu Tu[1], Samuel R Cramer[1], Nanyin Zhang[1,2,3,4]\***

[1]The Neuroscience Graduate Program, The Huck Institutes of the Life Sciences, Pennsylvania State University, University Park, United States; [2]Center for Neural Engineering, Pennsylvania State University, University Park, United States; [3]Center for Neurotechnology in Mental Health Research, Pennsylvania State University, University Park, United States; [4]Department of Biomedical Engineering, Pennsylvania State University, University Park, United States

**Abstract** Resting-state brain networks (RSNs) have been widely applied in health and disease, but the interpretation of RSNs in terms of the underlying neural activity is unclear. To address this fundamental question, we conducted simultaneous recordings of whole-brain resting-state functional magnetic resonance imaging (rsfMRI) and electrophysiology signals in two separate brain regions of rats. Our data reveal that for both recording sites, spatial maps derived from band-specific local field potential (LFP) power can account for up to 90% of the spatial variability in RSNs derived from rsfMRI signals. Surprisingly, the time series of LFP band power can only explain to a maximum of 35% of the temporal variance of the local rsfMRI time course from the same site. In addition, regressing out time series of LFP power from rsfMRI signals has minimal impact on the spatial patterns of rsfMRI-based RSNs. This disparity in the spatial and temporal relationships between resting-state electrophysiology and rsfMRI signals suggests that electrophysiological activity alone does not fully explain the effects observed in the rsfMRI signal, implying the existence of an rsfMRI component contributed by 'electrophysiology-invisible' signals. These findings offer a novel perspective on our understanding of RSN interpretation.

**\*For correspondence:**
nuz2@psu.edu

**Competing interest:** The authors declare that no competing interests exist.

## eLife assessment

This **important** study combines fMRI and electrophysiology in sedated and awake rats to show that LFPs strongly explain spatial correlations in resting-state fMRI but only weakly explain temporal variability. The authors propose that other, electrophysiology-invisible mechanisms contribute to the fMRI signal. The evidence supporting the separation of spatial and temporal correlations is **convincing**, and the authors consider alternative potential factors that could account for the differences in spatial and temporal correlation that were observed. This work will be of interest to researchers who study the mechanisms behind resting-state fMRI.

## Introduction

Sophisticated brain function requires coordinated activities from separate brain regions, collectively forming functional brain networks. Functional brain networks in humans and animals are predominantly studied using the method of resting-state functional magnetic resonance imaging (rsfMRI), which measures the synchronization of brain-wide spontaneous blood-oxygen-level-dependent (BOLD) signals. These networks are commonly referred to as resting-state brain networks (RSNs).

**eLife digest** The brain contains many cells known as neurons that send and receive messages in the form of electrical signals. The neurons in different regions of the brain must coordinate their activities to enable the brain to operate properly.

Researchers often use a method called resting-state functional magnetic resonance imaging (rsfMRI) to study how different areas of the brain work together. This method indirectly measures brain activity by detecting the changes in blood flow to different areas of the brain. Regions that are working together will become active (that is, have higher blood flow and corresponding rsfMRI signal) and inactive (have lower blood flow and a lower rsfMRI signal) at the same time. These coordinated patterns of brain activity are known as "resting-state brain networks" (RSNs).

Previous studies have identified RSNs in many different situations, but we still do not fully understand how these changes in blood flow are related to what is happening in the neurons themselves. To address this question, Tu et al. performed rsfMRI while also measuring the electrical activity (referred to as electrophysiology signals) in two distinct regions of the brains of rats. The team then used the data to generate maps of RSNs in those brain regions.

This revealed that rsfMRI signals and electrophysiology signals produced almost identical maps in terms of the locations of the RSNs. However, the electrophysiology signals only contributed a small amount to the changes in the local rsfMRI signals over time at the same recording site. This suggests that RSNs may arise from cell activities that are not detectable by electrophysiology but do regulate blood flow to neurons.

The findings of Tu et al. offer a new perspective for interpreting how rsfMRI signals relate to the activities of neurons. Further work is needed to explore all the features of the electrophysiology signals and test other methods to compare these features with rsfMRI signals in the same locations.

Despite the widespread application of BOLD-derived RSNs in both health and disease contexts, their relationship to the underlying neural activity remains incompletely understood. This is a fundamental issue, as the BOLD signal is known to indirectly reflect neural activity through accompanying hemodynamic and metabolic changes, a mechanism known as neurovascular coupling (NVC). While tight NVC has been repeatedly demonstrated when neural activities are evoked by explicit external stimulation (*Goense and Logothetis, 2008*; *Logothetis et al., 2001*; *Nemoto et al., 2004*), this relationship in the resting state remains elusive. There is considerable evidence indicating that the spatial patterns of most RSNs effectively mirror established functional systems and activation patterns observed during various tasks (*Biswal et al., 1995*; *Glasser et al., 2016*; *Hampson et al., 2002*; *Lowe et al., 1998*; *Smith et al., 2009*), as well as patterns of brain structural networks (*Andrews-Hanna et al., 2010*; *Greicius et al., 2009*; *Lowe et al., 2008*). In addition, alterations in RSNs have been documented in several brain disorders, aligning with neuropathophysiological changes (*Buckner et al., 2009*). This collective body of evidence suggests a robust neural basis for RSNs. However, it is notable that despite these findings, the predictive power of resting-state electrophysiological signals for corresponding rsfMRI time series is often relatively low (*Mateo et al., 2017*; *Schölvinck et al., 2010*; *Winder et al., 2017*), and some studies have even demonstrated a disconnection between neural activity and hemodynamic signals under specific conditions (*Maier et al., 2008*; *Zhang et al., 2019*). Moreover, various studies have reported divergent electrophysiological correlates of the rsfMRI signal across a broad spectrum of LFP bands, spanning from infraslow signals (*Hiltunen et al., 2014*; *Pan et al., 2013*) and low-frequency delta/sub-delta band signals (*He et al., 2008*; *Lu et al., 2007*) to high-frequency gamma-band signals (*Bastos et al., 2015*; *Foster et al., 2015*; *Keller et al., 2013*; *Mukamel et al., 2005*; *Nir et al., 2008*; *Magri et al., 2012*), as well as spiking activity (*Mukamel et al., 2005*). These findings suggest that the rsfMRI signal may reflect diverse aspects of neural activity. Taken together, how RSNs and rsfMRI relate to spontaneous neural activity remains unclear (*Laufs et al., 2003*; *Liu et al., 2011*; *Mantini et al., 2007*), highlighting a significant gap in our understanding of functional brain networks.

To address this critical issue, we systematically investigated the role of electrophysiological activity in determining specific spatiotemporal patterns of BOLD-based RSNs. In conjunction with whole-brain rsfMRI, we simultaneously recorded electrophysiology signals in the primary motor cortex (M1) and

anterior cingulate cortex (ACC) in rats under light sedation and wakefulness. These brain regions were chosen due to their distinct roles in sensorimotor function and integrative cognition, respectively. Our data show that in both light-sedation and awake states, the spatial patterns of RSNs derived from gamma-band power closely resemble BOLD-derived RSNs for both the M1 and ACC, and lower-frequency band-derived RSNs exhibit inversed spatial patterns, both indicating strong neural underpinning of RSNs. However, the temporal profiles of band-limited LFP powers at both recording sites exhibit considerably lower temporal correlations with the corresponding local BOLD time courses. Moreover, regressing out the gamma-band power or powers of all LFP bands has only limited effects on the spatial patterns of BOLD-derived RSNs, collectively suggesting LFP powers contribute only partially to the local rsfMRI signal. This disparity in spatial and temporal relationships between resting-state BOLD and electrophysiology signals implies that there might be an electrophysiology-invisible component of brain activity that significantly influences the rsfMRI signal and RSNs.

## Results

To systematically analyze the spatiotemporal relationship between resting-state electrophysiology and fMRI signals, we conducted simultaneous recordings of whole-brain rsfMRI and electrophysiology signals in the M1 and ACC in rats under both light-sedation (combination of low-dose dexmedetomidine [initial bolus of 0.05 mg/kg followed by a constant infusion at the rate of 0.1 mg/kg/hr] and low-dose isoflurane [0.3%]) and awake states (*Figure 1A*). The accuracy of electrode placement in the M1 and ACC was confirmed using T2-weighted structural images (*Figure 1—figure supplement 1*). Raw electrophysiology data were initially preprocessed to remove MR artifacts using a template regression approach (*Tu and Zhang, 2022*). Subsequently, the LFP was extracted by bandpass filtering preprocessed electrophysiology data within the frequency range of 0.1–300 Hz. An illustration of denoised LFP is depicted in *Figure 1A* and *Figure 1—figure supplement 2*. Band-specific LFP power was computed using a conventional LFP band definition (delta: 1–4 Hz, theta: 4–7 Hz, alpha: 7–13 Hz, beta: 13–30 Hz, gamma: 40–100 Hz). *Figure 1B and C* illustrates the cross correlations between LFP power and BOLD signal in the M1 across the LFP spectrum (*Figure 1B*, 1 Hz band interval) and for individual LFP bands (*Figure 1C*). These data demonstrate that gamma-band power is positively correlated with the BOLD signal, while lower-frequency bands display negative peak correlations with the BOLD signal. Additionally, the lag of the BOLD signal is approximately 2 s for all bands, consistent with the hemodynamic response function (HRF) delay previously reported in rodents (*Schölvinck et al., 2010*; *Winder et al., 2017*).

### LFP and rsfMRI signals derive consistent RSN spatial patterns in lightly sedated rats

We first examined the spatial relationship between brain-wide rsfMRI signals and frequency band-specific LFP powers in lightly sedated rats. For each recording site, its BOLD-derived RSN was obtained as the seedmap, calculated by voxel-wise correlating the regionally averaged BOLD time series of the seed (M1 in *Figure 1F* and ACC in *Figure 2B*) with BOLD time series of individual brain voxels. This seedmap conventionally represents the resting-state functional connectivity (RSFC) pattern for the seed region. To assess the extent to which this RSN could be obtained using the LFP signal recorded from the same location (M1 or ACC), we convolved the power of each LFP band with a rodent-specific HRF (*Figure 1D*, *Tong et al., 2019*) to generate the LFP band-predicted BOLD signal. Subsequently, we voxel-wise correlated this signal with the brain-wide rsfMRI signal, producing the LFP band-derived RSN map (*Figure 1E*, *Figure 2A*). Our findings revealed that the gamma-band power-derived RSN map exhibited high spatial consistency with the corresponding BOLD-derived RSN map (i.e. seedmap). Specifically, for M1, the voxel-to-voxel Pearson correlation coefficient (CC) between the mean gamma-derived RSN map (*Figure 1E*) and mean BOLD-derived RSN map (*Figure 1F*) was 0.95 (*Figure 1K*, $R^2=0.90$), indicating 90% of the variance in the M1 BOLD-derived RSN map could be explained by the gamma-derived map. Conversely, spatial maps generated by lower-frequency bands displayed inverse correlations with the M1 BOLD-derived RSN map, with a trend of increasingly negative spatial CC in lower-frequency bands (*Figure 1E–J*, delta: CC = –0.78; theta: CC = –0.78; alpha: CC = –0.5; beta: CC = –0.34), consistent with the LFP-BOLD cross correlations for these bands shown in *Figure 1B and C*. These relationships are repeatable in the ACC (*Figure 2A–G*, delta: CC = –0.62;

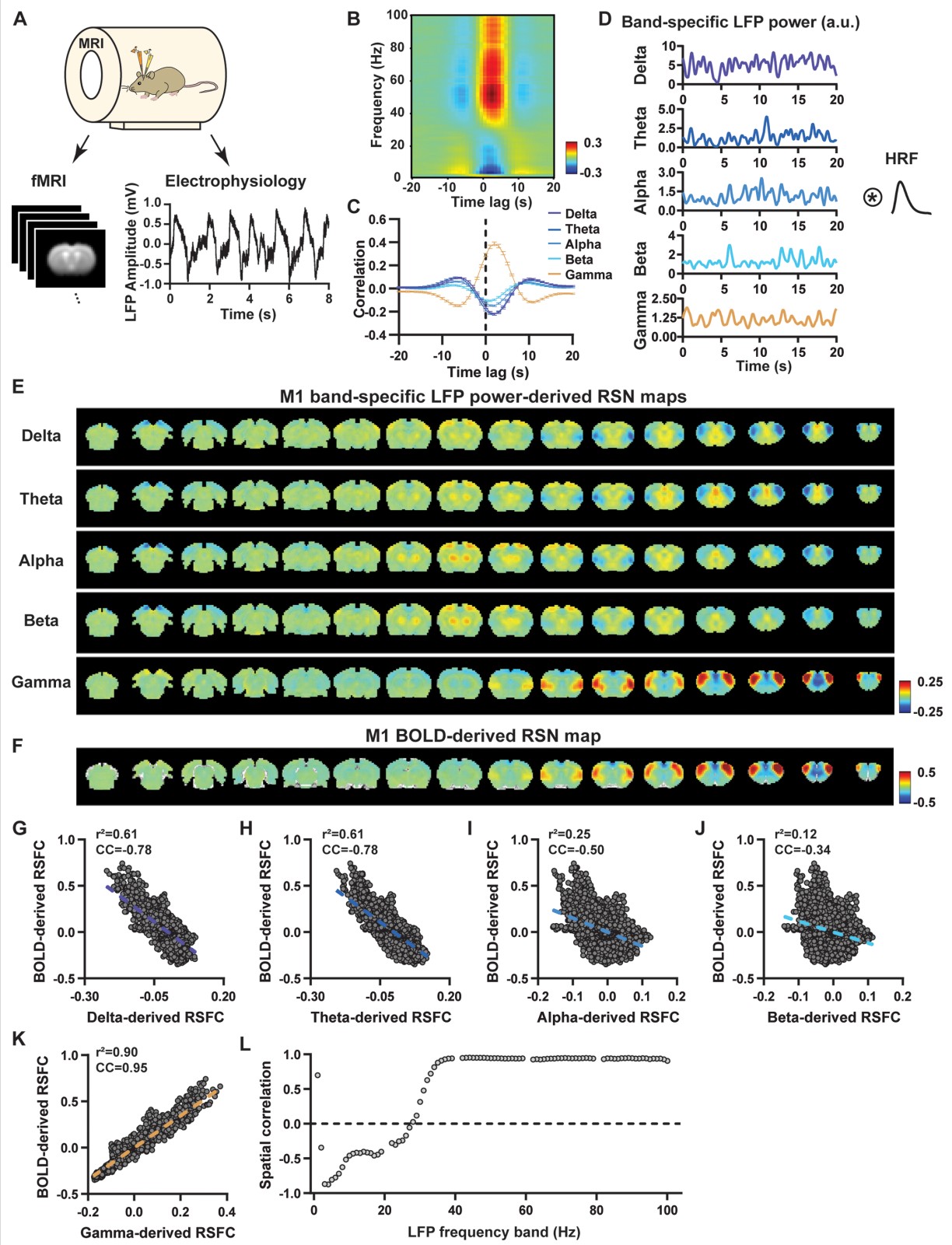

**Figure 1.** Highly consistent resting-state brain network (RSN) spatial patterns derived from local field potential (LFP) and resting-state functional magnetic resonance imaging (rsfMRI) signals in the primary motor cortex (M1) of lightly sedated rats. (**A**) Simultaneous acquisition setup for whole-brain rsfMRI and electrophysiology signals in the M1 and anterior cingulate cortex (ACC). (**B**) Cross correlations between the rsfMRI signal and LFP power in the M1 across the frequency range of 0.1–100 Hz (band interval: 1 Hz; lag range: –20 to 20 s). (**C**) Cross correlations between the rsfMRI signal

*Figure 1 continued on next page*

*Figure 1 continued*

and powers of individual LFP bands in the M1. Error bars: SEM. (**D**) Exemplar powers of individual LFP bands. Convolving these powers with a rodent-specific hemodynamic response function (HRF) generates the corresponding LFP-predicted blood-oxygen-level-dependent (BOLD) signals. (**E**) M1 band-specific LFP power-derived RSN maps, obtained by voxel-wise correlating the LFP-predicted BOLD signal for each band with BOLD signals of all brain voxels. (**F**) M1 BOLD-derived RSN map (i.e. M1 seedmap), obtained by voxel-wise correlating the regionally averaged BOLD time course of the seed (i.e. **M1**) with BOLD time courses of all brain voxels. (**G–K**) Spatial similarity between the M1 BOLD-derived RSN map and the M1 LFP-derived RSN map for each band, quantified by their voxel-to-voxel spatial correlations (G: delta, CC = –0.78; H: theta, CC = –0.78; I: alpha, CC = –0.5; J: beta, CC = –0.34; K: gamma, CC = 0.95). (**L**) Spatial correlations between the M1 BOLD-derived RSN map and RSN maps derived by individual 1-Hz bands across the full LFP spectrum.

The online version of this article includes the following figure supplement(s) for figure 1:

**Figure supplement 1.** Representative T2-weighted structural images confirming the electrode location in (**A**) primary motor cortex (M1) and (**B**) anterior cingulate cortex (ACC).

**Figure supplement 2.** Exemplar denoised local field potential (LFP) signal.

**Figure supplement 3.** Spatial relationship between primary motor cortex (M1) local field potential (LFP)-derived and M1 blood-oxygen-level-dependent (BOLD)-derived resting-state brain networks (RSNs) without global signal regression in resting-state functional magnetic resonance imaging (rsfMRI) data preprocessing.

**Figure supplement 4.** Spatial correlations of primary motor cortex (M1) local field potential (LFP)-derived and blood-oxygen-level-dependent (BOLD)-derived resting-state brain networks (RSNs) in awake animals.

**Figure supplement 5.** Anatomical connectivity labeled by tracers.

theta: CC = –0.3; alpha: CC = –0.12; beta: CC = 0.12; gamma: CC = 0.85). These results suggest that the spatial patterns of BOLD-based RSNs can be reliably obtained using band-specific LFP signals.

To confirm that these findings were not an artifact of specific frequency cutoffs we adopted for any LFP band, we repeated the analysis for all individual 1-Hz bands across the full LFP spectrum. Once again, we observed a gradual transition from negative to positive spatial correlations in LFP-derived RSN maps with the corresponding BOLD-derived RSN maps as the LFP signal changed from low to high frequencies (*Figure 1L* for M1; *Figure 2H* for ACC). Additionally, as a control analysis, we temporally shuffled the gamma-band power in the ACC, convolved it with the HRF, and recalculated the correlation map (*Figure 2—figure supplement 1*). As a result of this manipulation, the spatial pattern observed in *Figure 2A* disappeared, suggesting that the observed LFP-derived spatial patterns were specifically related to the LFP signal, rather than an artifact of the HRF. We also confirmed that all our results are not sensitive to the rsfMRI data preprocessing step of global signal regression (*Figure 1—figure supplement 3*, *Figure 2—figure supplement 2*).

In summary, our data collectively indicate that BOLD-derived RSNs can be reliably replicated using LFP power from the same site in lightly sedated rats, underscoring the critical involvement of neural activity in RSN spatial patterns.

## Temporal correlation between LFP power and local rsfMRI signal is significant but considerably weaker

Given the high reliability of the gamma power in determining spatial patterns of BOLD-based RSNs, it is logical to expect the HRF-convolved gamma power should reliably predict the rsfMRI time series from the same location. To test this hypothesis, we calculated temporal correlations between local rsfMRI time series and HRF-convolved LFP powers for individual scans at each recording site, and then averaged the resulting correlation values across scans. Surprisingly, we found that the local rsfMRI signal exhibited considerably weaker temporal correlations with LFP powers. In the M1, the LFP-BOLD temporal correlations gradually shifted from negative to positive as the LFP signal transitioned from low to high frequencies, mirroring the trend observed in spatial correlations (*Figure 1E–K*). However, the absolute magnitude of these CCs was considerably lower, despite that they were all statistically significant (one-sample t-tests, delta: CC = –0.20, $p=2.1 \times 10^{-57}$; theta: CC = –0.19, $p=7.1 \times 10^{-62}$; alpha: CC = –0.11, $p=1.2 \times 10^{-42}$; beta: CC = –0.06, $p=8.4 \times 10^{-15}$; gamma: CC = 0.37; $p=2.1 \times 10^{-58}$; number of scans = 159). Similar results were observed in the ACC (one-sample t-tests; delta: CC = –0.13, $p=1.9 \times 10^{-28}$; theta: CC = –0.04, $p=8.4 \times 10^{-7}$; alpha: CC = –0.03, $p=6.0 \times 10^{-5}$; beta: CC = –0.01, p=0.1; gamma: CC = 0.18, $p=6.7 \times 10^{-42}$; number of scans = 172). Additionally, we confirmed that lower temporal correlations are not due to the HRF used (*Figure 3—figure supplement 1*).

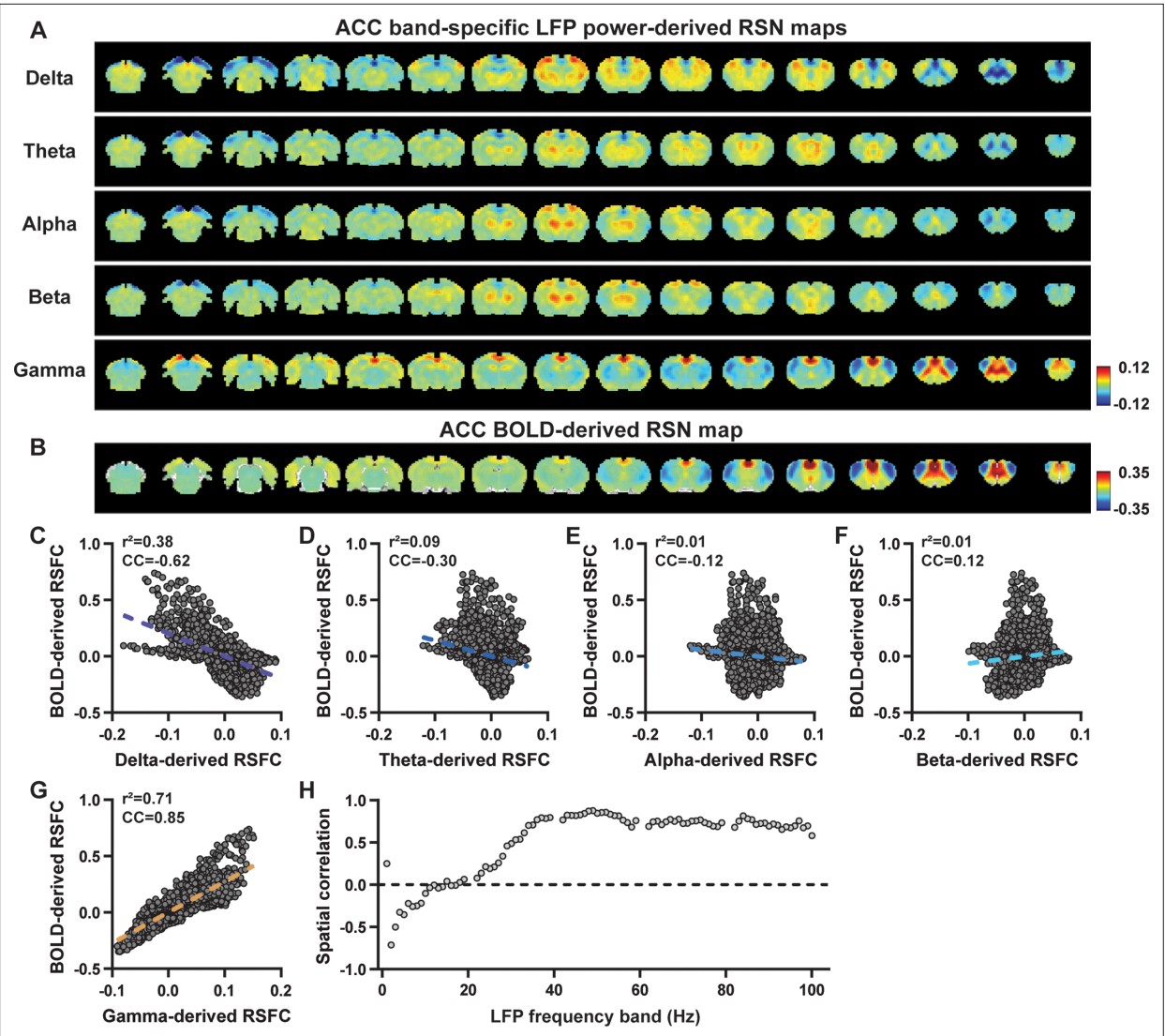

**Figure 2.** Highly consistent resting-state brain network (RSN) spatial patterns derived from local field potential (LFP) and resting-state functional magnetic resonance imaging (rsfMRI) signals in the anterior cingulate cortex (ACC) of lightly sedated rats. (**A**) ACC band-specific LFP power-derived RSN maps, obtained by voxel-wise correlating the LFP-predicted blood-oxygen-level-dependent (BOLD) signal for each band with BOLD signals of all brain voxels. (**B**) ACC BOLD-derived RSN map (i.e. ACC seedmap), obtained by voxel-wise correlating the regionally averaged BOLD time course of the seed (i.e. ACC) with BOLD time courses of all brain voxels. (**C–G**) Spatial similarity between the ACC BOLD-derived RSN map and the ACC LFP-derived RSN map for each band, quantified by their voxel-to-voxel spatial correlations (C: delta, CC = –0.62; D: theta, CC = –0.30; E: alpha, CC = –0.12; F: beta, CC = 0.12; G: gamma, CC = 0.85). (**H**) Spatial correlations between the ACC BOLD-derived RSN map and RSN maps derived by individual 1-Hz bands across the full LFP spectrum.

The online version of this article includes the following figure supplement(s) for figure 2:

**Figure supplement 1.** Control analysis for local field potential (LFP)-derived spatial pattern.

**Figure supplement 2.** Spatial relationship between anterior cingulate cortex (ACC) local field potential (LFP)-derived and ACC blood-oxygen-level-dependent (BOLD)-derived resting-state brain networks (RSNs) without global signal regression in resting-state functional magnetic resonance imaging (rsfMRI) data preprocessing.

**Figure supplement 3.** Spatial correlations of anterior cingulate cortex (ACC) local field potential (LFP)-derived and blood-oxygen-level-dependent (BOLD)-derived resting-state brain networks (RSNs) in awake animals.

A notable difference in our calculation of spatial and temporal correlations may contribute to the disparity in their CC values. When computing spatial correlations, we first generated LFP- and BOLD-derived RSN maps for each scan, and then averaged these maps within each group before calculating spatial correlations using the averaged maps (*Figures 1 and 2*). Conversely, for temporal correlations,

we initially computed for CCs for individual scans, and then averaged the resulting correlation values across scans. This approach was chosen as averaging time courses first would diminish the actual signal due to the semi-random nature of spontaneous brain activities. Consequently, the variance in averaged RSN spatial maps might be lower than the variance in time series of individual scans, which can result in higher apparent spatial correlations than temporal correlations. To control this factor, we also computed spatial correlations between LFP- and BOLD-derived RSN maps for individual scans, and then averaged the corresponding correlations across scans. Similar to scan-wise temporal correlations, scan-wise spatial correlations were significant for all LFP bands (one-sample t-tests; in the M1, delta: CC = –0.37, p=4.9 × $10^{-52}$; theta: CC = –0.39, p=3.9 × $10^{-62}$; alpha: CC = –0.24, p=5.8 × $10^{-37}$; beta: CC = –0.15, p=3.9 × $10^{-15}$; gamma: CC = 0.58, p=1.0 × $10^{-56}$; in the ACC, delta: CC = –0.26, p=2.6 × $10^{-27}$; theta: CC = –0.09, p=1.7 × $10^{-6}$; alpha: CC = –0.04, p=0.04; beta: CC = 0.03, p=0.06; gamma: CC = 0.33; p=5.4 × $10^{-40}$). Comparisons of scan-wise spatial and temporal correlations (*Figure 3A and B*) indicate that even after controlling for the variance level, the magnitude of spatial correlations remains appreciably higher than that of temporal correlations for all bands (paired t-tests across individual scans; in the M1, delta: p=1.01 × $10^{-35}$; theta: p=3.74 × $10^{-50}$; alpha: p=3.54 × $10^{-25}$; beta: p=1.18 × $10^{-12}$; gamma: p=7.02 × $10^{-42}$; in the ACC, delta: p=6.73 × $10^{-21}$; theta: p=5.75 × $10^{-5}$; alpha: p=0.74; beta: p=7.34 × $10^{-5}$; gamma: p=3.43 × $10^{-29}$).

Given that the gamma power-derived RSN map can explain ~90% of the spatial variance of the BOLD-derived RSN map (*Figure 1K*, R=0.95, $R^2$=0.90) when noise is diminished by averaging RSN maps across scans, we ask how much variance of the local BOLD time series the gamma power can explain without significant influence of noise. Although we cannot directly average rsfMRI/electrophysiology time courses across scans to reduce noise levels, we can estimate the true LFP-BOLD temporal correlation by quantitatively evaluating the effect of noise on correlation values. To achieve this aim, we utilized the difference between the spatial correlation of averaged M1 RSN maps (i.e. referred to as denoised data, R=0.95, *Figure 1K*) and that of unaveraged RSN maps (i.e. referred to as with-noise data, R=0.58, *Figure 3A*, gamma-band power). Using this difference, we simulated two fixed signals with a true correlation of 0.95. By introducing varying levels of noise to the signals, we determined at what noise level the apparent correlation between the two signals became 0.58 (*Figure 3C*). Specifically, in each trial, random noise at a defined contrast-to-noise ratio (CNR) level was added to the simulated signals, and this process was repeated 159 times (i.e. equal to the number of scans in our study) for a given CNR level. At each CNR level, the correlation was calculated based on either the averaged signals from all 159 trials (i.e. simulating denoised data, *Figure 3C*), or the signals of individual trials (i.e. simulating with-noise data) before averaging resulting correlations across trials.

As anticipated, lower CNR values correspond to lower apparent trial-wise correlation values. Interestingly, we discovered that the trial-wise apparent correlation of 0.58, with the true correlation of 0.95, corresponds to the CNR of 1.3 (*Figure 3D and E*), which aligns with the CNR of BOLD contrast reported in the literature (*Atkinson et al., 2008*). At this CNR level, we estimated that the true BOLD-LFP temporal correlation in the M1 should be approximately 0.59 ($R^2$=0.35, *Figure 3F and G*), when the apparent correlation is 0.37 as measured by the gamma-BOLD temporal correlation in our real data (*Figure 3A*). These findings indicate the temporal information provided by gamma power can only explain a minor portion (approximately 35%) of the temporal variance in the BOLD time series, even after accounting for the noise effect, which is in line with the reported correlation values between the cerebral blood volume (CBV) and fluctuations in GCaMP signal in head-fixed mice during periods of immobility (R=0.63) (*Ma et al., 2016*).These results are also consistent with previous reports of relatively weak temporal correlations between gamma power and hemodynamic signals at rest obtained using different imaging modalities (*Schölvinck et al., 2010*; *Winder et al., 2017*). Furthermore, our simulation suggests that the difference in the number of data points (1200 in temporal correlation calculation vs. 6157 in spatial correlation calculation) has a negligible influence on correlation values (*Figure 3D–G*).

## Regressing out LFP powers has limited impact on RSN spatial patterns

Given the lower predictive value of LFP power on the local rsfMRI signal, we investigated the extent to which the temporal information of LFP powers affects the RSN spatial patterns. The gamma-band power in the M1 (or ACC), after convolving with HRF, was linearly regressed out from rsfMRI signals of all brain voxels. As expected, the spatial patterns of gamma power-derived RSN maps observed

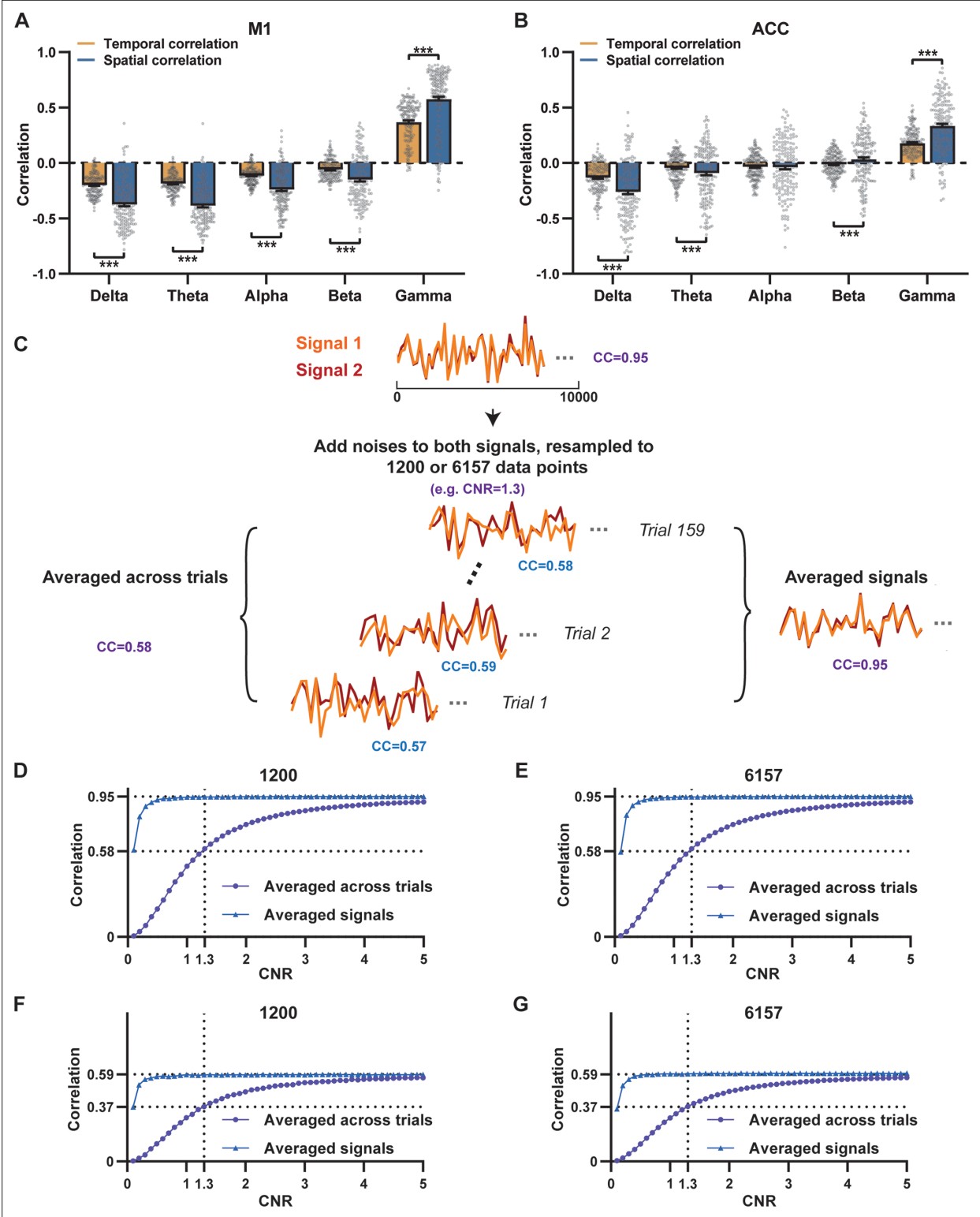

**Figure 3.** Disparity in spatial and temporal correlations persists after controlling for the noise effect. (**A,B**) Comparison of scan-wise spatial and temporal correlations (paired t-tests across individual scans. ***: p<0.005). (**C–G**) Simulation to evaluate factors affecting apparent correlation values including contrast-to-noise ratio (CNR) and the number of data points. (**C**) Two fixed signals with a true correlation coefficient (CC) of 0.95 are simulated (10,000 data points) with random noise added at a given CNR level. This process is repeated 159 times (i.e. the number of scans in our study) for each CNR level. At each CNR, the CC is calculated based on either the averaged signals from all 159 trials (i.e. denoised data, triangle dots in D–G), or signals of individual trials (i.e. with-noise data, round dots in D–G) before averaging the resulting correlations across trials. (**D**) Simulated signals resampled

*Figure 3 continued on next page*

*Figure 3 continued*

to 1200 data points (equal to the number of time points used to calculate temporal correlations). (**E**) Simulated signals resampled to 6157 data points (equal to the number of brain voxels used to calculate spatial correlations). Importantly, we can replicate the difference between true (R=0.95) and apparent (R=0.58) correlations obtained from denoised data and with-noise data, respectively, when CNR = 1.3. Therefore, we estimate that the CNR of our blood-oxygen-level-dependent (BOLD) data is ~1.3. (**F,G**) The same process as C–E with the true correlation of 0.59. This true correlation value is obtained by iteratively setting different true correlation values and searching for the one that provides the trial-wise apparent correlation of 0.37, as measured by the gamma-BOLD temporal correlation in our real data shown in (**A**), at CNR = 1.3. (**F**) Simulated signals resampled to 1200 data points. (**G**) Simulated signals resampled to 6157 data points.

The online version of this article includes the following figure supplement(s) for figure 3:

**Figure supplement 1.** Blood-oxygen-level-dependent (BOLD)-gamma power correlations in the two-dimensional space of hemodynamic response function (HRF) parameters.

**Figure supplement 2.** Comparison of scan-wise spatial and temporal correlations for (**A**) primary motor cortex (M1) and (**B**) anterior cingulate cortex (ACC) in awake rats.

**Figure supplement 3.** Disparity in spatial and temporal relationships between resting-state local field potential (LFP) and blood-oxygen-level-dependent (BOLD) signals quantified using alternative methods.

in *Figures 1E and 2A* disappeared after the regression (*Figures 4A and 5A*). However, this regression process minimally altered M1/ACC BOLD-derived RSN maps (*Figures 4B and 5B*). This result remained consistent when the powers of all LFP bands were voxel-wise regressed out from rsfMRI signals using soft regression (*Figures 4C and 5C*). Soft regression was utilized to address the multicollinearity issue in the regression model resulting from potential correlations between LFP bands, as this method allows only unique components in five LFP bands to be regressed out.

To investigate whether the regression process is disproportionately dominated by time points with the largest LFP amplitude (i.e. outliers), we recalculated the M1/ACC BOLD-derived RSN maps after removing rsfMRI volumes corresponding to peaks in the M1/ACC gamma power (i.e. time points with the signal amplitude above the 85th percentile in HRF-convolved gamma power of each scan, *Figures 4D, E and 5D*). The spatial similarities between the BOLD-derived RSN maps before and after gamma power regression, all band power regression, or peak removal are summarized in *Figures 4F and 5E*, showing that the removal of gamma power has limited impact on the M1/ACC RSN maps.

To control for the potential nonlinear relationship between band-specific LFP powers and the rsfMRI signal, we calculated the mutual information between band-limited LFP powers and rsfMRI signals for all brain voxels (*Figure 5—figure supplement 1*). The results show limited mutual information between any band-specific power and voxel-wise rsfMRI signals, indicating that the nonlinear component between BOLD and electrophysiological signals, reflected by mutual information, does not significantly influence RSN spatial patterns, which is consistent with the report that macroscopic resting-state brain dynamics are best described by linear models (*Nozari et al., 2024*). These data collectively indicate that the temporal fluctuations of LFP have limited effects on BOLD-derived RSN spatial patterns.

## Disparity in temporal and spatial correlations persists across different physiological states

To determine whether the disparity between temporal and spatial correlations of resting-state LFP and fMRI signals we observed is a specific phenomenon under anesthesia or can be generalized to different physiological states, we repeated the experiment in awake rats. Despite that the physiological dynamics are substantially different, we still found higher spatial correlations between LFP-derived maps (*Figure 1—figure supplement 4A*) and the BOLD-derived RSN map (*Figure 1—figure supplement 4B*) in the M1. Also similar to what we showed in anesthetized rats (*Figures 1 and 2*), CCs gradually changed from negative in low-frequency bands to positive in high-frequency bands, which were revealed both in conventionally defined bands (*Figure 1—figure supplement 4C–G*) and 1 Hz bands (*Figure 1—figure supplement 4H*). In the ACC, while spatial correlations in low-frequency bands were somewhat diminished, the overall pattern remained similar (*Figure 2—figure supplement 3*). Similar to the results in *Figure 3*, significant but weaker temporal correlations between the rsfMRI signal and HRF-convolved gamma-band power were observed in awake rats (one-sample t-tests; for M1, delta: CC = –0.05, p=2.3 × 10⁻⁴; theta: CC = –0.06, p=4.7 × 10⁻⁶; alpha: CC = –0.06, p=4.6 × 10⁻⁵; beta: CC = –0.032, p=3.0 × 10⁻³; gamma: CC = 0.04, p=0.02; for ACC, delta: CC = 0.02,

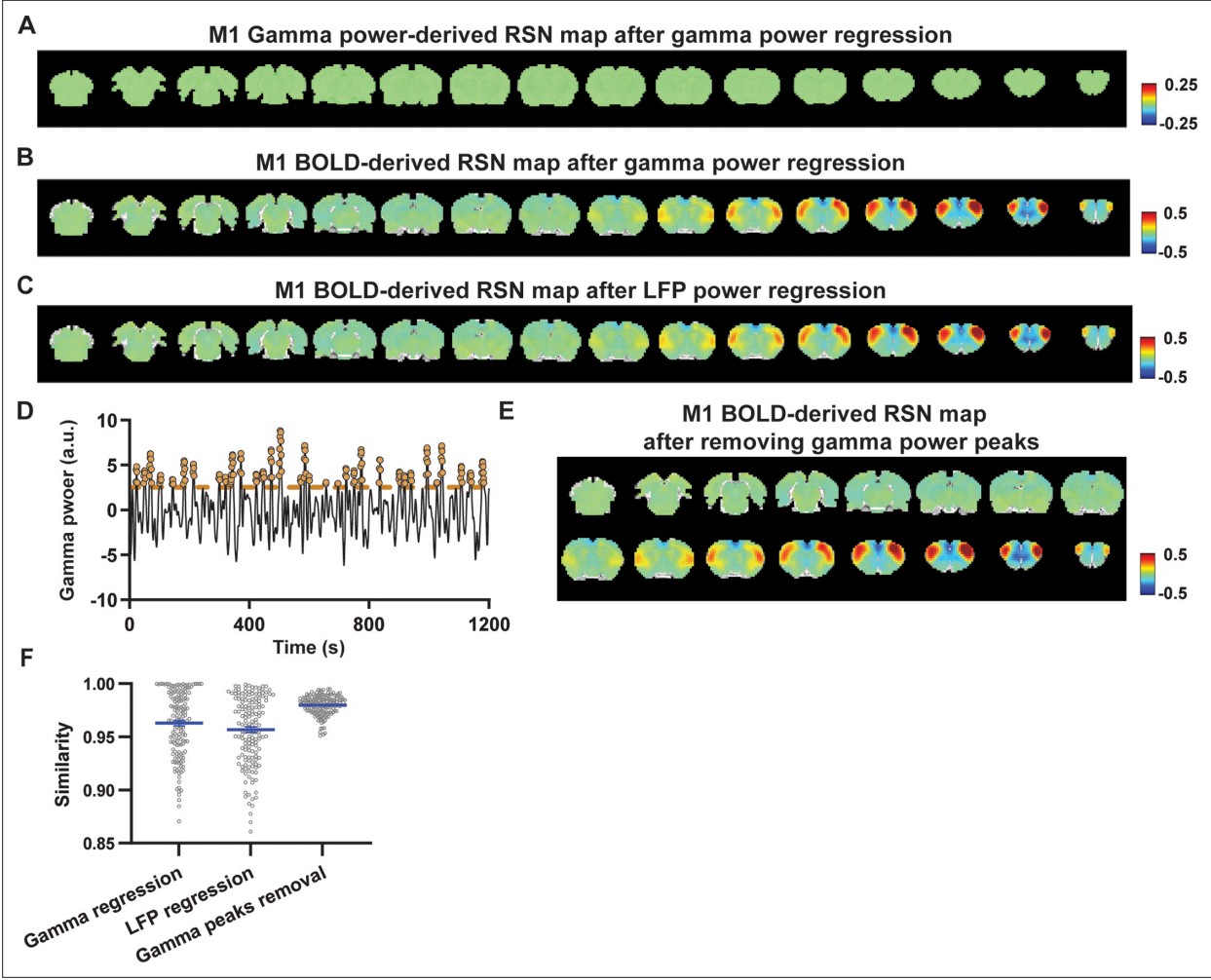

**Figure 4.** Impact of removing the electrophysiology signal on primary motor cortex (M1) blood-oxygen-level-dependent (BOLD)-derived resting-state brain network (RSN) spatial patterns. (**A**) RSN map derived from gamma power after regressing out the hemodynamic response function (HRF)-convolved gamma power in the M1 from resting-state functional magnetic resonance imaging (rsfMRI) signals of all brain voxels. (**B**) M1 BOLD-derived RSN map (i.e. M1 seedmap) after voxel-wise regression of the HRF-convolved gamma power from rsfMRI signals. (**C**) M1 BOLD-derived RSN map after voxel-wise regression of all five local field potential (LFP) band powers from rsfMRI signals using soft regression. (**D**) Peaks of HRF-convolved gamma power in one representative scan. (**E**) M1 BOLD-derived RSN map after removing 15% of rsfMRI time points corresponding to gamma peaks. (**F**) Spatial similarity of M1 BOLD-derived RSN maps before and after gamma power regression, regression of all LFP band powers, or gamma peak removal.

The online version of this article includes the following figure supplement(s) for figure 4:

**Figure supplement 1.** Impact of removing electrophysiology signals on primary motor cortex (M1) and anterior cingulate cortex (ACC) resting-state brain network (RSN) spatial patterns in awake animals.

p=0.29; theta: CC = 0.0009, p=0.96; alpha: CC = –0.02, p=0.31; beta: CC = 0.006, p=0.66; gamma: CC = 0.10, p=$9.52 \times 10^{-7}$; number of scans = 50). Scan-wise comparison between temporal and spatial correlations is shown in *Figure 3—figure supplement 2*. Overall, both spatial and temporal correlations showed lower magnitudes in the awake state, likely due to increased variability from motion and other physiological fluctuations, as well as the smaller number of scans compared to the light-sedation state (50 awake scans, 159 light-sedation scans for M1, and 172 light-sedation scans for ACC). Nonetheless, the consistent pattern of lower temporal but higher spatial correlations between gamma power and the rsfMRI signal supports the notion that this disparity is a general phenomenon across different physiological states.

The lack of significant alteration in BOLD-derived RSN maps after regressing out gamma-band power or powers of all LFP bands in both ACC and M1 of awake rats (*Figure 4—figure supplement 1*) further reinforces our earlier findings. These results indicate that the major findings observed in

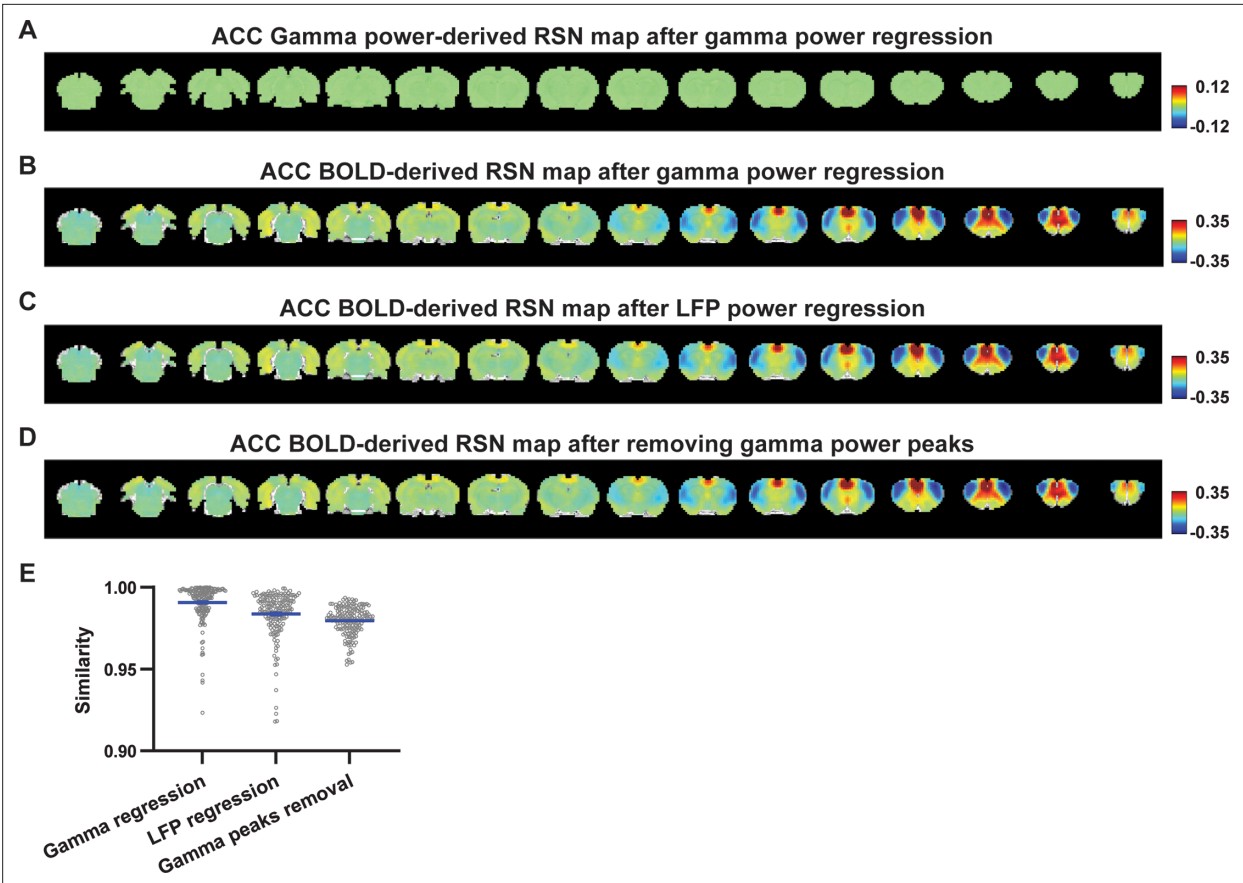

**Figure 5.** Impact of removing the electrophysiology signal on anterior cingulate cortex (ACC) blood-oxygen-level-dependent (BOLD)-derived resting-state brain network (RSN) spatial patterns. (**A**) RSN map derived from gamma power after regressing out the hemodynamic response function (HRF)-convolved gamma power in the ACC from resting-state functional magnetic resonance imaging (rsfMRI) signals of all brain voxels. (**B**) ACC BOLD-derived RSN map (i.e. ACC seedmap) after voxel-wise regression of the HRF-convolved gamma power from rsfMRI signals. (**C**) ACC BOLD-derived RSN map after voxel-wise regression of all five local field potential (LFP) band powers from rsfMRI signals using soft regression. (**D**) ACC BOLD-derived RSN map after removing 15% of rsfMRI time points corresponding to gamma peaks. (**E**) Spatial similarity of ACC BOLD-derived RSN maps before and after gamma power regression, regression of all LFP band powers, or gamma peak removal.

The online version of this article includes the following figure supplement(s) for figure 5:

**Figure supplement 1.** Mutual information between anterior cingulate cortex (ACC) band-limited local field potential (LFP) powers and brain-wide resting-state functional magnetic resonance imaging (rsfMRI) signals.

lightly anesthetized rats, including the disparity between temporal and spatial correlations, can be replicated in unanesthetized rats. Therefore, it appears that these results are not specific to the effects of anesthesia but rather reflect fundamental aspects of the relationship between electrophysiological and hemodynamic signals in the brain across different physiological states.

## Ongoing rsfMRI signal could be contributed by electrophysiology-invisible brain activities

Our findings indicate that the LFP signal can capture RSN patterns that account for nearly all the spatial variance observed in BOLD-based seedmaps. However, the temporal dynamics of the LFP signal only explain a minor fraction of the local BOLD time series and have minimal impact on the spatial patterns of BOLD-based RSNs. To reconcile this apparent contradiction, we propose a theoretical model, described as follows.

Brain activity consists of components measurable by electrophysiology and others that are electrophysiology-invisible. In addition to the electrophysiology activity, electrophysiology-invisible brain activities, such as those involving nNOS neurons and astrocytes, actively contribute to NVC and can exert a significant influence on the rsfMRI signal. During the resting state, these two components

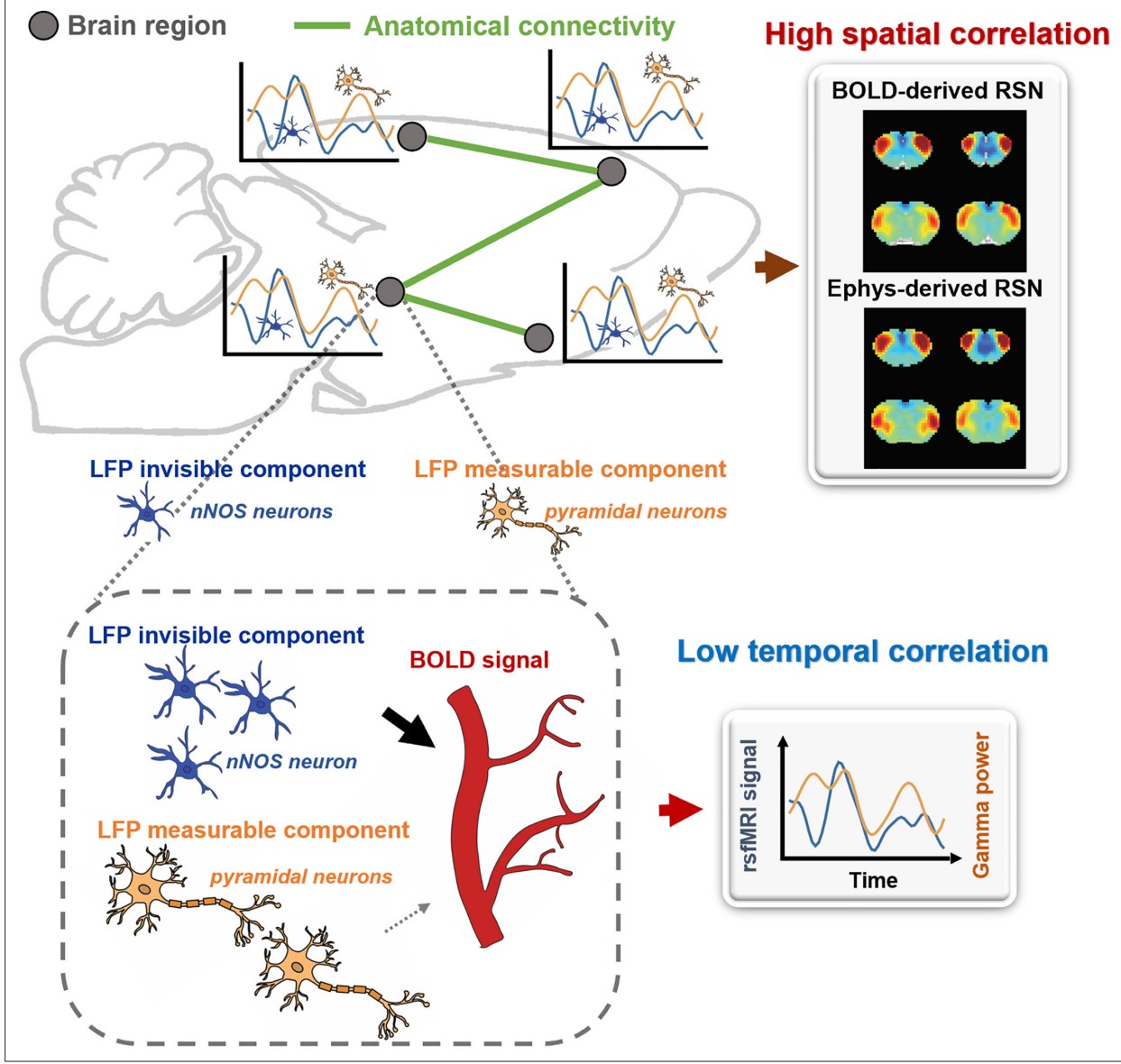

**Figure 6.** A theoretic model that can explain the disparity in spatial and temporal correlations between resting-state electrophysiology and functional magnetic resonance imaging (fMRI) signals.

The online version of this article includes the following figure supplement(s) for figure 6:

**Figure supplement 1.** Schematic diagram illustrating that distal neuromodulation, which have strong vasoactive effects, can contribute to low temporal correlations between electrophysiology and resting-state functional magnetic resonance imaging (rsfMRI) signals.

may not be synchronized, lowering temporal correlations between electrophysiology and rsfMRI signals. Another factor contributing to low LFP-BOLD temporal correlations could be neuromodulations from distal modulator nuclei (e.g. locus coerulues and/or basal forebrain), which exert strong vasoactive effects but may not proportionately affect electrophysiology activity. Moreover, the signaling of electrophysiology activities and that of electrophysiology-invisible are both constrained by the same anatomical pathways. This allows the two signals to generate similar RSN spatial patterns in parallel measured by the BOLD signal, which can reflect both direct and indirect anatomical connectivity. The model is summarized in *Figure 6* and *Figure 6—figure supplement 1*, and further details will be discussed in the next section.

## Discussion

BOLD-derived RSNs have been widely investigated in multiple species (*Damoiseaux et al., 2006*; *Lu et al., 2012*; *Mantini et al., 2011*; *Liang et al., 2011*) and have been applied in various physiological and pathological conditions. However, the neural mechanisms underlying RSNs and the rsfMRI signal remain incompletely understood. To shed more light on this issue, we conducted systematic analysis of simultaneously recorded electrophysiology and rsfMRI signals in both lightly anesthetized and awake animals. Our findings reveal that both electrophysiology and rsfMRI signals can generate highly consistent brain-wide RSN patterns. However, the temporal information of the LFP signal contributes only minimally to the local BOLD time series at the same recording site. These seemingly paradoxical findings, along with those reported in the literature, can potentially be reconciled by the theoretical model we propose, which suggests that RSNs may also arise from electrophysiology-invisible brain activities that play a significant role in NVC. Together, our data and model offer a new perspective for interpreting the neural basis underlying the resting-state BOLD signal.

The spatial correspondence between BOLD- and electrophysiology-derived RSNs has been repeatedly reported across various physiological states and species using different methods. Studies employing electroencephalography or electrocorticography in humans have shown that RSNs derived from the power of multiple-site electrophysiological signals exhibit similar spatial patterns to classic BOLD-derived RSNs, such as the default-mode network (*Hacker et al., 2017*; *Kucyi et al., 2018*). This high spatial correspondence between rsfMRI and LFP signals can even be found at the columnar level (*Shi et al., 2017*). Similarly, simultaneous recordings of resting-state calcium and fMRI signals in awake rats have revealed highly consistent spatial patterns between calcium- and BOLD-associated RSNs (*Ma et al., 2022*). Moreover, voltage-sensitive dye imaging in mice has unveiled comparable sensory-evoked and hemisphere-wide activity motifs represented in spontaneous activity in both lightly anesthetized and awake states (*Mohajerani et al., 2013*). Furthermore, in a more recent study by Vafaii and colleagues, overlapping cortical networks were identified using both fMRI and calcium imaging modalities, suggesting that networks observable in fMRI studies exhibit corresponding neural activity spatial patterns (*Vafaii et al., 2024*). These results align well with the notion that RSN spatial patterns are highly consistent with known functional systems and activation patterns observed in task-based studies (*Biswal et al., 1995*; *Glasser et al., 2016*; *Hampson et al., 2002*; *Lowe et al., 1998*; *Smith et al., 2009*), as well as patterns of structural networks (*Andrews-Hanna et al., 2010*; *Greicius et al., 2009*; *Lowe et al., 2008*). Taken together, previous studies and our data indicate that both electrophysiology and rsfMRI measurements can generate consistent RSN spatial patterns, strongly suggesting that the spatial structures of RSNs are dictated by neural activities.

Previous studies have also indicated that RSNs are likely constrained by axonal projections. For instance, consistent sensory-evoked and hemisphere-wide activity motifs in mice, as revealed using voltage-sensitive dye imaging, are defined by regional axonal projections (*Mohajerani et al., 2013*). Our previous work in awake rats further demonstrate spatial consistency between RSNs and anatomical connectivity patterns in thalamocortical networks (*Liang et al., 2013*). In the current study, we compared RSNs of M1 and ACC to their anatomical networks defined by the axonal projection patterns obtained from the Allen Brain Institute database (*Figure 1—figure supplement 5*, *Oh et al., 2014*), illustrating that RSNs revealed by either the BOLD or LFP signal closely resemble the corresponding anatomical networks. It is important to note that RSNs measured by functional connectivity can reflect both direct and indirect connectivity and thus may not necessarily have identical spatial patterns as the corresponding anatomical networks (*Honey et al., 2009*).

In contrast to the seemingly strong LFP-BOLD relationship inferred from their high spatial correlations, we observed appreciably lower (yet significant) temporal correlations between the two signals from the same recording sites (M1 and ACC). Regressing out LFP powers has limited impact on RSN spatial patterns, reinforcing the notion that the contribution of temporal variations of the LFP signal to RSN spatial patterns is minor. These results are supported by previous research demonstrating weak but significant correlations between CBV changes and spontaneous gamma-band LFP or multiunit activity in awake, head-fixed mice (*Winder et al., 2017*). Importantly, persistent spontaneous fluctuations in CBV were observed even after blocking local neural spiking and glutamatergic input, as well as noradrenergic receptors, indicating that hemodynamic signal fluctuations may not dominantly reflect local ongoing electrophysiology activity (*Winder et al., 2017*). Comparably low temporal correlations between gamma-band power and the rsfMRI signal have been reported in monkeys (*Schölvinck et al.,*

*2010*; *Shmuel and Leopold, 2008*). Additionally, data from our group show similarly low temporal correlations between spontaneous calcium peaks, a measure of neural spiking activity, and the rsfMRI signal in awake rats (*Ma et al., 2022*). Furthermore, Vafaii et al. revealed notable differences in functional connectivity strength measured by fMRI and calcium imaging, despite an overlapping spatial pattern of cortical networks identified by both modalities (*Vafaii et al., 2024*). These findings collectively suggest that while the temporal correlation between electrophysiology and rsfMRI signals is significant, the effect size of this correlation might be small. This result appears to remain consistent even for infraslow LFP activity (<1 Hz). Our data show that in the M1, the temporal correlation between infraslow LFP power and the rsfMRI signal was 0.08, while both derived consistent RSN spatial patterns (spatial correlation = 0.7), in line with the report that RSNs can be derived from infraslow LFP activity (*Li et al., 2022*). It is noteworthy that these results differ from two earlier studies in isoflurane-anesthetized rats, which found the LFP power in the primary somatosensory cortex was highly correlated with BOLD fluctuations (*Liu et al., 2011*; *Pan et al., 2011*). This discrepancy may be attributed to brain-wide synchronization during burst suppression in deeply anesthetized states in those studies.

Why would electrophysiology and rsfMRI signals exhibit unmatched spatial and temporal correlations? Our model offers one possible scenario that can reconcile this discrepancy. It hypothesizes that a portion of the rsfMRI signal is driven by electrophysiology-invisible brain activities involved in NVC. LFP records various neural activities, such as synaptic potentials and voltage-gated membrane fluctuations, reflecting the input and local neural processing of a particular brain region. However, electrophysiology cannot measure activities from certain cell populations, while electrophysiology-invisible components can trigger vasoactive responses and significantly contribute to the rsfMRI signal. For instance, the electrical activity of nNOS neurons is not detectable by electrophysiology because the nNOS neuron population is very small, yet it strongly contributes to NVC. Chemogenetic or pharmacological stimulation of nNOS neurons causes vasodilation without detectable changes in LFP (*Echagarruga et al., 2020*). In addition, astrocytes, a type of glia cell, coordinate communication between neurons and blood vessels and play a crucial role in NVC. Astrocytes regulate vessel tone by releasing signaling molecules such as ATP, arachidonic acid metabolites, and nitric oxide (*Iadecola and Nedergaard, 2007*; *Mulligan and MacVicar, 2004*; *Murphy et al., 1993*). Furthermore, astrocytes can alter the diameter of blood vessels by extending or retracting the endfeet wrapping around them (*Mills et al., 2022*; *Niu et al., 2019*). Optogenetic stimulation of astrocytes in transgenic mice without affecting neurons elicited a BOLD response, indicating that astrocyte activity alone can cause changes in the BOLD signal, independent of neuronal activity (*Takata et al., 2018*). Additionally, Uhlirova et al. conducted a study where they utilized optogenetic stimulation and two-photon imaging to investigate how the activation of different neuron types affects blood vessels in mice. They discovered that only the activation of inhibitory neurons led to vessel constriction, albeit with a negligible impact on LFP (*Uhlirova et al., 2016*). These studies collectively suggest electrophysiology-invisible activities can significantly drive the rsfMRI signal. Exclusively identifying all electrophysiology-invisible sources contributing to the rsfMRI signal is beyond the scope of this work. However, a key point is that as the resting-state LFP and electrophysiology-invisible (and thus rsfMRI) signals reflect different components of brain activities, they can be minimally synchronized and display low temporal correlations. Another possible factor that can contribute to low BOLD-LFP temporal correlations is direct modulation of the vasculature from distant modulator nuclei (e.g. locus coerulues and/or basal forebrain). Some neuromodulators, such as norepinephrine (*Bekar et al., 2012*; *Kim et al., 2016*) and acetylcholine (*Sato and Sato, 1995*), have vasoconstrictive/vasodilatory effects that are spatially targeted in distributed brain regions, and thus can modulate brain-wide rsfMRI signals. However, these neural modulation effects may not be proportionately reflected from the electrophysiology signal, lowering BOLD-LFP temporal correlations (see a schematic diagram in *Figure 6—figure supplement 1*). On the other hand, as the signaling of LFP- and electrophysiology-invisible components in functional networks is constrained by the same anatomical connectivity structure (*Liu et al., 2018b*; *Liu et al., 2018a*), the LFP- and BOLD-derived RSN spatial patterns can be highly similar and thus have high spatial correlations.

Our model can also potentially explain high spatial and temporal correlations when brain activation is evoked by external stimulation (*Winder et al., 2017*). At the evoked state, both LFP (and spiking activity) and electrophysiology-invisible components are temporally modulated by the same external

stimulation paradigm, leading to both high temporal and high spatial correlations between fMRI and electrophysiology signals.

## Potential pitfalls

Our proposed theoretic model represents just one potential explanation for the apparent discrepancy in temporal and spatial relationships between resting-state electrophysiology and BOLD signals. It is important to acknowledge that there may be other scenarios where a stronger temporal relationship between LFP and BOLD signals could manifest. For instance, recent research suggests that the relationship between LFP and rsfMRI signals may vary across different modes or instances (*Cabral et al., 2023*), which can be masked by correlations across the entire time series. Moreover, the 1-s temporal resolution employed in our study may obscure certain temporal correlations between LFPs and rsfMRI signals. Future investigations employing ultrafast fMRI imaging coupled with dynamic connectivity analysis could offer a more nuanced exploration of BOLD-LFP temporal correlations at higher temporal resolutions (*Bolt et al., 2022*; *Thompson et al., 2014*; *Ma and Zhang, 2018*).

In addition to LFP, various other features can be derived from electrophysiology signals and alternative methods for comparing electrophysiology and rsfMRI signals, such as rank correlation, warrant consideration. It is plausible that employing different features or comparison methods could yield a stronger BOLD-electrophysiology temporal relationship (*Ma et al., 2016*). In our current study we focused solely on band-limited LFP power as the primary feature in our analysis, given its prevalence in prior studies of LFP-rsfMRI correlates. More importantly, we demonstrate that band-specific LFP powers can yield spatial patterns nearly identical to those derived from rsfMRI signals, prompting a closer examination of the temporal relationship between these same features. Furthermore, since correlational analysis was used in studying the LFP-BOLD spatial relationship, we used the same analysis method when comparing their temporal relationship. Some other comparison methods such as rank correlation and transformation prior to comparison were also tested and results remain persistent (*Figure 3—figure supplement 3*). These findings align with the notion that, compared to nonlinear models, linear models offer superior predictive value for the rsfMRI signal using LFP data, as comprehensively illustrated in *Nozari et al., 2024* (also see *Figure 5—figure supplement 1*). Importantly, in this study, the predictive powers (represented by $R^2$) of various comparison methods tested all remain below 0.5 (*Nozari et al., 2024*), suggesting that while certain models may enhance the temporal relationship between LFP and BOLD signals, the improvement is likely modest. Further exploration involving the extraction of all possible features from electrophysiology signals and their examination in relation to the rsfMRI signal, as well as the exploration of alternative methods for comparing LFP and rsfMRI signals warrants more detailed analysis in future studies.

## Summary

Our study demonstrates that BOLD-based RSNs can be reliably derived by the electrophysiology signal. Nonetheless, the weak BOLD-LFP temporal correlations suggest that the dominant contributors to these networks might be signals not captured by electrophysiology. This finding provides a novel interpretation of RSNs. Importantly, this new concept of RSN signaling does not in any way diminish the importance of BOLD-based RSNs or the rsfMRI method. In fact, it makes fMRI even more important than previously thought because it might provide a new signal that traditional electrophysiology measures cannot provide.

## Materials and methods

### Animals

All experiments in the present study were approved by and conducted in accordance with guidelines from the Pennsylvania State University Institutional Animal Care and Use Committee (IACUC, protocol #: PRAMS201343583). Adult male Long-Evans rats weighing 300–500 g were obtained from Charles River Laboratory (Wilmington, MA, USA). Animals were housed in Plexiglas cages with food and water given ad libitum. The ambient temperature was maintained at 22–24°C under a 12 hr light:12 hr dark cycle.

### Surgery

MR-compatible electrodes were implanted in animals with aseptic stereotaxic surgeries. The rat was briefly anesthetized with isoflurane before receiving intramuscular injections of ketamine (40 mg/kg)

and xylazine (12 mg/kg). Baytril (2.5 mg/kg) and long-acting buprenorphine (1.0 mg/kg) were administered subcutaneously as antibiotics and analgesics, respectively. The animal was then endotracheally incubated and ventilated with oxygen using the PhysioSuite system (Kent Scientific Corporation). Body temperature was monitored and maintained at 37°C with a warming pad placed underneath the animal (PhysioSuite, Kent Scientific Corporation). Heart rate and $SpO_2$ were continuously monitored using a pulse oximetry (MouseSTAT Jr, Kent Scientific Corporation) throughout the surgery. After performing craniotomies over the right ACC (coordinates: anterior/posterior +1.5, medial/lateral +0.5, dorsal/ventral –2.8) and the left M1 (coordinates: anterior/posterior +3.2, medial/lateral –3, dorsal/ventral –2.8), two MR-compatible electrodes (MRCM16LP, NeuroNexus Inc) were carefully implanted into the ACC and M1, respectively. The reference and grounding wires from each electrode were wired together and connected to one of the two silver wires placed in the cerebellum. This electrode, which is a silicon-based micromachined probe, is capable of recording the LFP activity within a broad frequency range, starting from 0.1 Hz. At last, the skull was sealed with dental cement. After surgery, the animal was returned to the homecage and allowed to recover for at least 1 week before any experiment.

## Acclimation for awake imaging

Rats were restrained using a custom-designed restrainer during awake imaging sessions. To minimize stress and motion during the imaging process, animals underwent a 7-day acclimation procedure to the restrainer as well as the MRI environment and scanning noise. The duration of the acclimation procedure was gradually increased from 15 min on the first day to 60 min on days 4–7 days (i.e. 15 min on day 1, 30 min on day 2, 45 min on day 3, and 60 min on days 4–7). More details of the acclimation procedure can be found in previous publications from our laboratory (*Liang et al., 2012*) and other research groups (*Bergmann et al., 2016*; *Chang et al., 2016*).

## Simultaneous rsfMRI and electrophysiology recordings

All rsfMRI experiments were conducted on a 7T Bruker 70/30 BioSpec system running ParaVision 6.0.1 (Bruker, Billerica, MA, USA) using a homemade single loop surface coil at the *high field MRI facility* at the Pennsylvania State University. During each fMRI session, T2*-weighted rsfMRI images covering the entire brain were obtained using an echo planar imaging sequence with the following parameters: repetition time (TR)=1 s; echo time (TE)=15 ms; field of view = 3.2 × 3.2 cm$^2$; matrix size = 64 × 64; slice number = 20; slice thickness = 1 mm; volume number = 1200. Five to ten scans were repeated within each session. T2-weighted anatomical images were also acquired using a rapid acquisition with a relaxation enhancement sequence with the following parameters: TR = 3000 ms; TE = 40 ms; field of view = 3.2 × 3.2 cm$^2$; matrix size = 256 × 256; slice number = 20; slice thickness = 1 mm; repetition number = 6.

Six rats with two electrodes implanted in the ACC and M1 were imaged in both awake and lightly sedated states in separate fMRI sessions. Two additional rats with an electrode only implanted in the ACC were imaged in the light-sedation state. For both states, animals were restrained throughout the imaging session. In the light-sedation state, the animal was sedated with the combination of low-dose dexmedetomidine (initial bolus of 0.05 mg/kg followed by a constant infusion at the rate of 0.1 mg/kg/hr) and low-dose isoflurane (0.3%). Artificial tears were applied to protect the animal's eyes from drying out. Body temperature was maintained at 37°C using warm air and was monitored using a rectal thermometer.

Before imaging, the implanted electrodes were connected to MR-compatible LP16CH headstages and a PZ5 neurodigitizer amplifier (Tucker Davis Technologies (TDT) Inc, Alachua, FL, USA). Electrophysiology recording began 10 min before rsfMRI data acquisition and continued until the end of the imaging session using a TDT recording system and an RZ2 BioAmp Processor (TDT Inc, Alachua, FL, USA). The raw, unfiltered electrophysiology signal was sampled at 24,414 Hz and stored using the TDT Synapse software on a WS8 workstation.

## rsfMRI and electrophysiology data preprocessing

All data preprocessing and analysis were performed using MATLAB (Mathworks, Natick, MA, USA). First, the movement of each rsfMRI volume was estimated using the frame-wise displacement (FD). For the awake imaging data, volumes with FD >0.1 mm and their adjacent preceding and following

volumes were removed. If >25% of volumes in a scan were scrubbed, the entire scan was excluded from further analysis. For rsfMRI data collected in the lightly sedated state, scans with any volume that had FD >0.1 mm were removed from further analysis. Subsequently, data were preprocessed using the following steps: co-registration to a defined atlas, motion correction (SPM12), spatial smoothing using a Gaussian kernel (FWHM = 0.75 mm), voxel-wise nuisance regression with the regressors of motion parameters as well as signals from the white matter and ventricles, and the global brain signal, and, lastly, bandpass temporal filtering (0.01–0.1 Hz).

Raw electrophysiology data were preprocessed to remove the MR interference using a template regression method as previously described (*Tu and Zhang, 2022*). Briefly, the raw electrophysiology signal for each scan was first aligned to the corresponding rsfMRI scan, and segmented for each imaging slice based on the starting time of the scan. Next, an MRI interference template for each rsfMRI slice acquisition was obtained by averaging the electrophysiology data across all slices from all rsfMRI volumes. The template was further aligned to the electrophysiology data for each slice acquisition using cross correlation and was then linearly regressed out from the raw electrophysiology data. In addition, a series of notch filters for harmonics of the power supply (60 Hz and multiples of 60 Hz) and slice acquisition (20 Hz and multiples of 20 Hz) were applied to further denoise the data.

## Data analysis

The LFP power was obtained by bandpass filtering preprocessed electrophysiology data in the frequency range of 0.1–300 Hz. Based on the conventional LFP band definition (delta: 1–4 Hz, theta: 4–7 Hz, alpha: 7–13 Hz, beta: 13–30 Hz, gamma: 40–100 Hz) (*Lu et al., 2016*; *Zhang et al., 2020*), the LFP band power was computed using the MATLAB function *spectrogram* with a window size of 1 s and a step size of 0.1 s. To investigate the relationship between the LFP and rsfMRI signals, the time course of band-specific LFP power was convolved with an HRF (p = [4 4 1 1 6 0 32] for function *spm_hrf*) to generate the corresponding LFP-predicted BOLD signal. The HRF used was specific to rodents with a shorter onset time and time-to-peak as a faster HRF was reported in rats relative to humans (*Tong et al., 2019*). The temporal relationship between the LFP and fMRI signals was quantified using the Pearson correlation between the HRF-convolved LFP band power and the regionally averaged rsfMRI time course from voxels surrounding the implanted electrode for each site. To examine the potential impact of the HRF used, we calculated the BOLD-gamma power correlation using different HRFs with various response delays, ranging from 2 s to 8 s with the increment of 0.25 s, as well as different under-shoot delays ranging from 2 s to 12 s with the increment of 0.5 s (*Figure 3—figure supplement 1*).

The LFP-derived spatial correlation maps for the M1 and ACC were respectively generated by computing voxel-wise Pearson correlations between each HRF-convolved LFP band power and brain-wide rsfMRI signals. The seedmaps for the M1 and ACC were respectively obtained by calculating the voxel-wise Pearson correlations between the regionally averaged rsfMRI time course for each seed and rsfMRI signals of all brain voxels.

To determine the contribution of LFP powers to BOLD-based RSFC, we removed LFP powers from voxel-wise fMRI signals and then recalculated RSNs. Given that the gamma signal might be the most related to the rsfMRI signal, the time course of gamma-band power (convolved with HRF) was linearly regressed out from rsfMRI signals of all brain voxels. To examine the potential contributions of other LFP bands, all five LFP band powers (each convolved with HRF) were 'softly' removed from voxel-wise rsfMRI signals, meaning only the unique components in five bands were regressed out but the shared components were maintained. Specific details of soft regression can be found in *Griffanti et al., 2014*. This method can avoid 'over regression' when multiple regressors are involved particularly when regressors are correlated between themselves. Lastly, we removed rsfMRI volumes corresponding to peaks in the M1/ACC gamma power. The seedmaps of M1 and ACC were recalculated and compared before and after removing the LFP signal.

## Simulation

We simulated two fixed signals with the true Pearson correlation of 0.95. The first signal was generated using MATLAB function *rand* with 10,000 data points. The second signal with a defined correlation with the first signal (i.e. 0.95) was obtained based on the equation below:

$$B = A * Corr + \sqrt{1 - Corr^2} * N(0,1) \tag{1}$$

in which A represents the first signal, Corr is the desired Pearson CC, and N(0, 1) represents random values with the mean equal to 0 and standard deviation equal to 1. For each signal, random noise was added to achieve a CNR ranging from 0.1 to 5 with the step size of 0.1. CNR was quantified by the standard deviation of the signal over the standard deviation of the noise. This process was repeated 159 times (equal to the # of scans in the present study). The noise-added signals were resampled to either 1200 or 6157 data points, which corresponded to the total number of time points used to calculate temporal correlations and total number of brain voxels used to calculate spatial correlations, respectively, in our study. Pearson correlations between the resampled signals either based on the averaged signals from all 159 trials or on individual trials were calculated.

## Acknowledgements

We thank Dr. Patrick J Drew for his insightful scientific discussion. The present study was partially supported by National Institute of Neurological Disorders and Stroke (R01NS085200) and National Institute of Mental Health (RF1MH114224). The content is solely the responsibility of the authors and does not necessarily represent the official views of the National Institutes of Health.

## Additional information

### Funding

| Funder | Grant reference number | Author |
| --- | --- | --- |
| National Institute of Neurological Disorders and Stroke | R01NS085200 | Nanyin Zhang |
| National Institute of Mental Health | RF1MH114224 | Nanyin Zhang |

The funders had no role in study design, data collection and interpretation, or the decision to submit the work for publication.

### Author contributions

Wenyu Tu, Data curation, Formal analysis, Validation, Investigation, Visualization, Methodology, Writing – original draft; Samuel R Cramer, Resources, Data curation, Formal analysis, Validation, Investigation, Visualization, Methodology, Writing – original draft, Writing – review and editing; Nanyin Zhang, Conceptualization, Resources, Supervision, Funding acquisition, Validation, Investigation, Writing – original draft, Project administration, Writing – review and editing

### Author ORCIDs

Wenyu Tu http://orcid.org/0000-0003-3480-2098
Nanyin Zhang https://orcid.org/0000-0002-5824-9058

### Ethics

All experiments in the present study were approved by and conducted in accordance with guidelines from the Pennsylvania State University Institutional Animal Care and Use Committee (protocol #: PRAMS201343583).

Reviewer #1 (Public review): https://doi.org/10.7554/eLife.95680.3.sa1
Reviewer #2 (Public review): https://doi.org/10.7554/eLife.95680.3.sa2
Author response https://doi.org/10.7554/eLife.95680.3.sa3

## Additional files

### Supplementary files
• MDAR checklist

## Data availability

All preprocessed electrophysiology and fMRI data are deposited at https://zenodo.org/records/10537319.

The following dataset was generated:

| Author(s) | Year | Dataset title | Dataset URL | Database and Identifier |
|---|---|---|---|---|
| Tu W, Cramer SR, Zhang N | 2024 | Resting state fMRI and electrophysiology in rats | https://doi.org/10.5281/zenodo.10537319 | Zenodo, 10.5281/zenodo.10537319 |

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
