## [Editor Report · eLife assessment]

This **important** study combines fMRI and electrophysiology in sedated and awake rats to show that LFPs strongly explain spatial correlations in resting-state fMRI but only weakly explain temporal variability. The authors propose that other, electrophysiology-invisible mechanisms contribute to the fMRI signal. The evidence supporting the separation of spatial and temporal correlations is **convincing**, and the authors consider alternative potential factors that could account for the differences in spatial and temporal correlation that were observed. This work will be of interest to researchers who study the mechanisms behind resting-state fMRI.

---

## [Referee Report · Reviewer #1 (Public review)]

Tu et al investigated how LFPs recorded simultaneously with rsfMRI explain the spatiotemporal patterns of functional connectivity in sedated and awake rats. They find that connectivity maps generated from gamma band LFPs (from either area) explain very well the spatial correlations observed in rsfMRI signals, but that the temporal variance in rsfMRI data is more poorly explained by the same LFP signals. The authors excluded the effects of sedation in this effect by investigating rats in the awake state (a remarkable feat in the MRI scanner), where the findings generally replicate. The authors also performed a series of tests to assess multiple factors (including noise, outliers, etc., and nonlinearity of the data...) in their analysis.

This apparent paradox is then explained by a hypothetical model in which LFPs and neurovascular coupling are generated in some sense "in parallel" by different neuron types, some of which drive LFPs and are measured by ePhys, while others (nNOS, etc.) have an important role in neurovascular coupling but are less visible in Ephys data. Hence the discrepancy is explained by the spatial similarity of neural activity but the more "selective" LFPs picked up by Ephys account for the different temporal aspects observed.

This is a deep, outstanding study that harnesses multidisciplinary approaches (fMRI and ephys) for observing brain activity. The results are strongly supported by the comprehensive analyses done by the authors, that ruled out many potential sources for the observed findings. The study's impact is expected to be very large.

There are very few weaknesses in the work, but I'd point out that the 1-second temporal resolution may have masked significant temporal correlations between LFPs and spontaneous activity, for instance, as shown by Cabral et al Nature Communications 2023, and even in earlier QPP work from the Keilholz Lab. The synchronization of the LFPs may correlate more with one of these modes than the total signal. Perhaps a kind of "dynamic connectivity" analysis on the authors' data could test whether LFPs correlate better with the activity at specific intervals. However this could purely be discussed and left for future work, in my opinion.

---

## [Referee Report · Reviewer #2 (Public review)]

The authors investigate the disparity between spatial extant and temporal variance of electrophysiological-fMRI correlations in a rodent model. They found high correspondence in spatial extent but a disparity in temporal variance. From this, they propose a model of an electrophysiologically-invisible signal affecting temporal variance.

I remain skeptical about the "electrophysiologically invisible signal" model but the authors have done a much better job of both explaining it and hedging it in this version. Readers can decide for themselves.

The revision submitted by the authors substantially improves writing and methods.

---

## [Author Response]

The following is the authors’ response to the original reviews.

**eLife assessment**
This important study combines fMRI and electrophysiology in sedated and awake rats to show that LFPs strongly explain spatial correlations in resting-state fMRI but only weakly explain temporal variability. They propose that other, electrophysiology-invisible mechanisms contribute to the fMRI signal. The evidence supporting the separation of spatial and temporal correlations is convincing, however, the support of electrophysiological-invisible mechanisms is incomplete, considering alternative potential factors that could account for the differences in spatial and temporal correlation that were observed. This work will be of interest to researchers who study the fundamental mechanisms behind resting-state fMRI.

We appreciate the encouraging comments. We added a section in discussion that thoroughly discussed the potential alternative factors that could account for the differences in spatial and temporal correlation that we observed.

**Public Reviews:**

**Reviewer #1 (Public Review):**
Tu et al investigated how LFPs recorded simultaneously with rsfMRI explain the spatiotemporal patterns of functional connectivity in sedated and awake rats. They find that connectivity maps generated from gamma band LFPs (from either area) explain very well the spatial correlations observed in rsfMRI signals, but that the temporal variance in rsfMRI data is more poorly explained by the same LFP signals. The authors excluded the effects of sedation in this effect by investigating rats in the awake state (a remarkable feat in the MRI scanner), where the findings generally replicate. The authors also performed a series of tests to assess multiple factors (including noise, outliers, and nonlinearity of the data) in their analysis.This apparent paradox is then explained by a hypothetical model in which LFPs and neurovascular coupling are generated in some sense "in parallel" by different neuron types, some of which drive LFPs and are measured by ePhys, while others (nNOS, etc.) have an important role in neurovascular coupling but are less visible in Ephys data. Hence the discrepancy is explained by the spatial similarity of neural activity but the more "selective" LFPs picked up by Ephys account for the different temporal aspects observed.This is a deep, outstanding study that harnesses multidisciplinary approaches (fMRI and ephys) for observing brain activity. The results are strongly supported by the comprehensive analyses done by the authors, which ruled out many potential sources for the observed findings. The study's impact is expected to be very large.Comment: There are very few weaknesses in the work, but I'd point out that the 1second temporal resolution may have masked significant temporal correlations betweenLFPs and spontaneous activity, for instance, as shown by Cabral et al Nature Communications 2023, and even in earlier QPP work from the Keilholz Lab. The synchronization of the LFPs may correlate more with one of these modes than the total signal. Perhaps a kind of "dynamic connectivity" analysis on the authors' data could test whether LFPs correlate better with the activity at specific intervals. However, this could purely be discussed and left for future work, in my opinion.

We appreciate this great point. Indeed, it is likely that LFP and rsfMRI signals are more strongly related during some modes/instances than others, and hence correlation across the entire time series may have masked this effect. In addition, we agree that 1-second temporal resolution may obscure some temporal correlations between LFPs and rsfMRI signal. The choice of 1-second temporal resolution was made to be consistent with the TR in our fMRI experiment, considering the slow hemodynamic response. Ultrafast fMRI imaging combined with dynamic connectivity analysis in a future study might enable more detailed examination of BOLD-LFP temporal correlations at higher temporal resolutions. We have added the following paragraph to the revised manuscript:

“Our proposed theoretic model represents just one potential explanation for the apparent discrepancy in temporal and spatial relationships between resting-state electrophysiology and BOLD signals. It is important to acknowledge that there may be other scenarios where a stronger temporal relationship between LFP and BOLD signals could manifest. For instance, recent research suggests that the relationship between LFP and rsfMRI signals may vary across different modes or instances (Cabral et al., 2023), which can be masked by correlations across the entire time series. Moreover, the 1-second temporal resolution employed in our study may obscure certain temporal correlations between LFPs and rsfMRI signals. Future investigations employing ultrafast fMRI imaging coupled with dynamic connectivity analysis could offer a more nuanced exploration of BOLD-LFP temporal correlations at higher temporal resolutions (Bolt et al., 2022; Cabral et al., 2023; Ma and Zhang, 2018; Thompson et al., 2014).”

**Reviewer #2 (Public Review):**
The authors address a question that is interesting and important to the sub-field of rsfMRI that examines electrophysiological correlates of rsfMRI. That is, while electrophysiology-produced correlation maps often appear similar to correlation maps produced from BOLD alone (as has been shown in many papers) is this actually coming from the same source of variance, or independent but spatially-correlated sources of variance? To address this, the authors recorded LFP signals in 2 areas (M1 and ACC) and compared the maps produced by correlating BOLD with them to maps produced by BOLD-BOLD correlations. They then attempt to remove various sources of variance and see the results.The basic concept of the research is sound, though primarily of interest to the subset of rsfMRI researchers who use simultaneous electrophysiology. However, there are major problems in the writing, and also a major methodological problem.Major problems with writing:Comment 1: There is substantial literature on rats on site-specific LFP recording compared to rsfMRI, and much of it already examined removing part of the LFP and examining rsfMRI, or vice versa. The authors do not cover it and consider their work on signal removal more novel than it is.

We have added more literature studies to the revised manuscript. It is important to note that while there exists a substantial body of literature on site-specific LFP recording coupled with rsfMRI, our paper makes a significant contribution by unveiling the disparity in temporal and spatial relationships between resting-state electrophysiological and fMRI signals. This goes beyond mere reporting of spatial/temporal correlations. Furthermore, our exploration of the impact of removing LFP on rsfMRI spatial patterns constitutes one among several analyses employed to demonstrate that the temporal fluctuations of LFP minimally affect BOLD-derived RSN spatial patterns. We wish to clarify that our intention is not to claim this aspect of our work is more novel than similar analyses conducted in previous studies (we apologize if our original manuscript conveyed that impression). Rather, the novelty lies in the objective of this analysis, which is to elucidate the displarity in temporal and spatial relationships between resting-state electrophysiological and fMRI signals—a crucial issue that has not been thoroughly addressed previously.

Comment 2: The conclusion of the existence of an "electrophysiology-invisible signal" is far too broad considering the limited scope of this study. There are many factors that can be extracted from LFP that are not used in this study (envelope, phase, infraslow frequencies under 0.1Hz, estimated MUA, etc.) and there are many ways of comparing it to the rsfMRI data that are not done in this study (rank correlation, transformation prior to comparison, clustering prior to comparison, etc.). The one non-linear method used, mutual information, is low sensitivity and does not cover every possible nonlinear interaction. Mutual information is also dependent upon the number of bins selected in the data. Previous studies (see 1) have seen similar results where fMRI and LFP were not fully commensurate but did not need to draw such broad conclusions.

First we would like to clarify that the existence of "electrophysiologyinvisible signal" is not necessarily a conclusion of the present study, per se, as described by the reviewer. As we stated in our manuscript, it is a proposed theoretical model. We fully acknowledge that this model represents just one potential explanation for the apparent discrepancy in temporal and spatial relationships between resting-state electrophysiology and BOLD signals. It is important to acknowledge that there may be other scenarios where a stronger temporal relationship between LFP and BOLD signals could manifest. This issue has been further clarified in the revised manuscript (see the section of Potential pitfalls).

We agree with the reviewer that not all factors that can be extracted from LFP are examined. In our current study we focused solely on band-limited LFP power as the primary feature in our analysis, given its prevalence in prior studies of LFP-rsfMRI correlates. More importantly, we demonstrate that band-specific LFP powers can yield spatial patterns nearly identical to those derived from rsfMRI signals, prompting a closer examination of the temporal relationship between these same features. Furthermore, since correlational analysis was used in studying the LFP-BOLD spatial relationship, we used the same analysis method when comparing their temporal relationship.

Extracting all possible features from the electrophysiology signal and examining their relationship with the rsfMRI signal or exploring all other types of ways of comparing LFP and rsfMRI signals goes beyond the scope of the current study. However, to address the reviewer’s concern, we tried a couple of analysis methods suggested by the reviewer, and results remain persistent. Figure S14 shows the results from (A) the rank correlation and (B) z transformation prior to comparison. We added these new results to the revised manuscript.

Comment 3: The writing refers to the spatial extent of correlation with the LFP signal as "spatial variance." However, LFP was recorded from a very limited point and the variance in the correlation map does not necessarily reflect underlying electrophysiological spatial distributions (e.g. Yu et al. Nat Commun. 2023 Mar 24;14(1):1651.)

The reviewer accurately pointed out that in our paper, “spatial variance” refers to the spatial variance of BOLD correlates with the LFP signal. Our objective is to assess the extent to which this spatial variance, which is derived from the neural activity captured by LFP in the M1 or ACC, corresponds to the BOLD-derived spatial patterns from the same regions. We acknowledge that this spatial variance may differ from the spatial map obtained by multi-site electrophysiology recordings. Nevertheless, numerous studies have consistently reported a high spatial correspondence between BOLD- and electrophysiology-derived RSNs using various methodologies across different physiological states in both humans and animals. For instance, research employing electroencephalography (EEG) or electrocorticography (ECoG) in humans demonstrates that RSNs derived from the power of multiple-site electrophysiological signals exhibit similar spatial patterns to classic BOLD-derived RSNs such as the default-mode network (Hacker et al., 2017; Kucyi et al., 2018). These studies well agree with our findings. Notably, the reference paper cited by the reviewer studies brain-wide changes during transitions between awake and various sleep stages, which is quite different from the brain states examined in our study.

Major method problem:Comment 4: Correlating LFP to fMRI is correlating two biological signals, with unknown but presumably not uniform distributions. However, correlating CC results from correlation maps is comparing uniform distributions. This is not a fair comparison, especially considering that the noise added is also uniform as it was created with the rand() function in MATLAB.

This is a good point. We examined the distributions of both LFP powers and fMRI signals. They both seem to follow a normal distribution. Below shows distributions of the two signals from a random scan. In addition, z transformation prior to comparison generated the same results (Fig. S14).

**Author response image 1. sa3fig1:** Exemplar distributions of (A) the fMRI signal of M1, and (B) HRF-convolved LFP power in M1.

**Reviewer #1 (Recommendations For The Authors):**
Comment 1: In the Discussion, a few more calcium imaging papers could be fruitfully discussed (e.g. Ma et al Resting-state hemodynamics are spatiotemporally coupled to synchronized and symmetric neural activity in excitatory neurons, PNAS 2016, or more recently Vafaii et al, Multimodal measures of spontaneous brain activity reveal both common and divergent patterns of cortical functional organization, Nat Comms 2024).

We appreciate this suggestion. We have added the following discussions to the revised manuscript:

“These findings indicate the temporal information provided by gamma power can only explain a minor portion (approximately 35%) of the temporal variance in the BOLD time series, even after accounting for the noise effect, which is in line with the reported correlation value between the cerebral blood volume and fluctuations in GCaMP signal in head-fixed mice during periods of immobility (R = 0.63) (Ma et al., 2016).”

“It is plausible that employing different features or comparison methods could yield a stronger BOLD-electrophysiology temporal relationship (Ma et al., 2016).”

“Furthermore, in a more recent study by Vafaii and colleagues, overlapping cortical networks were identified using both fMRI and calcium imaging modalities, suggesting that networks observable in fMRI studies exhibit corresponding neural activity spatial patterns (Vafaii et al., 2024).”

“Furthermore, Vafaii et. al. revealed notable differences in functional connectivity strength measured by fMRI and calcium imaging, despite an overlapping spatial pattern of cortical networks identified by both modalities (Vafaii et al., 2024).”

Comment 2: Similarly when discussing the "invisible" populations, perhaps Uhlirova et al eLife 2016 should be mentioned as some types of inhibitory processes may also be less clearly observed in LFPs but rather strongly contribute to NVC.

We appreciate the suggestion. We added the following sentences to the revised manuscript.

“Additionally, Uhlirova et al. conducted a study where they utilized optogenetic stimulation and two-photon imaging to investigate how the activation of different neuron types affects blood vessels in mice. They discovered that only the activation of inhibitory neurons led to vessel constriction, albeit with a negligible impact on LFP (Uhlirova et al., 2016).”

**Reviewer #2 (Recommendations For The Authors):**
Major problems with writing:Comment 1: The authors need to review past work to better place their study in the context of the literature (some review articles: Lurie et al. Netw Neurosci. 2020 Feb 1;4(1):30-69. & Thompson et al. Neuroimage. 2018 Oct 15;180(Pt B):448-462.)Here are some LFP and BOLD "resting state" papers focused on dynamic changes.Many of these papers examine both spatial and temporal extents of correlations. Several of these papers use similar methods to the reviewed paper.Also, many of these papers dispute the claim that correlations seen are"electrophysiology invisible signal." Note that I am NOT saying that "electrophysiology invisible" correlations do not exist (it seems very likely some DO exist). However, the authors did not show that in the reviewed paper, and some of the correlations which they call an "electrophysiology invisible signal" probably would be visible if analyzed in a different manner.

Quite a few literature studies that the reviewer suggested were already included in the original manuscript. We have also added more literature studies to the revised manuscript. Again, we would like to emphasize that the novelty of our study centers on the discovery of the disparity in temporal and spatial relationships between resting-state electrophysiological and fMRI signals. See below our responses to individual literature studies listed.

In humans:https://pubmed.ncbi.nlm.nih.gov/38082179/ Predicts by using models the paper under review does not use here.

The following discussion was added to the revised manuscript:

“Some other comparison methods such as rank correlation and transformation prior to comparison were also tested and results remain persistent (Fig. S14). These findings align with the notion that, compared to nonlinear models, linear models offer superior predictive value for the rsfMRI signal using LFP data, as comprehensively illustrated in Nozari et al., 2024 (also see Fig. S7). Importantly, in this study, the predictive powers (represented by R2) of various comparison methods tested all remain below 0.5 (Nozari et al., 2024), suggesting that while certain models may enhance the temporal relationship between LFP and BOLD signals, the improvement is likely modest.”

In nonhuman primates: https://pubmed.ncbi.nlm.nih.gov/34923136/ Most of the variance that could be creating resting state networks is in the <1 Hz band which the paper under review did not study

We also examined infraslow LFP activity (< 1Hz) in our data. Consistent with the finding in the reference paper (Li et al., 2022), infraslow LFP power and the BOLD signal can derive consistent RSN spatial patterns (for M1, spatial correlation = 0.70), while the temporal correlation remains very low (temporal correlation = 0.08). These results and the reference paper were added to the revised manuscript.

https://pubmed.ncbi.nlm.nih.gov/28461461/ Compares actual spread of LFP vs. spread of BOLD instead of just correlation between LFP and BOLD.

The following sentence has been added to the revised manuscript.

“This high spatial correspondence between rsfMRI and LFP signals can even be found at the columnar level (Shi et al., 2017).”

https://pubmed.ncbi.nlm.nih.gov/24048850/ Comparison of small (from LFP) to large (from BOLD) spatial correlations in the context of temporal correlations.

In this study, researchers compared neurophysiological maps and fMRI maps of the inferior temporal cortex in macaques in response to visual images. They observed that the spatial correlation increased as the neurophysiological maps got greater levels of spatial smoothing. This suggests that fMRI can capture large-scale spatial information, but it may be limited in capturing fine details. Although interesting, this paper did not study the electrophysiology-fMRI relationship at the resting state and hence is not very relevant to our study.

https://pubmed.ncbi.nlm.nih.gov/20439733/ Electrophysiology from a single site can correlate across nearly the entire cerebral cortex.

We have included the discussion of this paper in the original manuscript.

https://pubmed.ncbi.nlm.nih.gov/18465799/ The original dynamic BOLD and LFP work from 2008 by Shmuel and Leopold included spatiotemporal dynamics.

We have included the discussion of this paper in the original manuscript.

In rodents:https://pubmed.ncbi.nlm.nih.gov/34296178/ Better electrophysiological correspondence was found using alternate methods the paper under review does not use.

This study investigates the electrophysiological correspondence in taskbased fMRI, while our study focused on resting state signals.

https://pubmed.ncbi.nlm.nih.gov/31785420/ Electrophysiological basis of co-activation patterns, similar comparisons to the paper under review.

We have included the discussion of this paper in the original manuscript.

https://pubmed.ncbi.nlm.nih.gov/29161352/ Cross-frequency coupling of LFP modulating the BOLD, perhaps more so than raw amplitudes.

This paper investigated the impact of AMPA microinjections in the VTA and found reduced ventral striatal functional connectivity, correlation between the delta band and BOLD signal, and phase–amplitude coupling of low-frequency LFP and highfrequency LFP, suggesting changes in low-frequency LFP might modulate the BOLD signal.

Consistent with our study, we also found that low-frequency LFP is negatively coupled with the BOLD signal, but we did not investigate changes in neurovascular coupling with disturbed neural activity using pharmacological methods, and hence, we did not discuss this paper in our study.

https://pubmed.ncbi.nlm.nih.gov/24071524/ This paper did the same kind of tests comparing LFP-BOLD correlations to BOLD-BOLD correlations as the paper under review.

This study examined the neural mechanism underpinning dynamic restingstate fMRI, revealing a spatiotemporal coupling of infra-slow neural activity with a quasiperiodic pattern (QPP). While our current investigation centered on stationary restingstate functional connectivity, we acknowledge that dynamic analysis will provide additional value for investigating the relationship between LFP and rsfMRI signals. This warrants more investigation in a future study. This point has been added to the revised manuscript.

https://pubmed.ncbi.nlm.nih.gov/24904325/ This paper found that different frequencies of electrophysiology (including ones not studied in the reviewed paper) contribute independently to the BOLD signal

This paper identified phase-amplitude coupling in rats anesthetized with isoflurane but not with dexmedetomidine, indicating that this coupling arises from a special type of neural activity pattern, burst-suppression, which was probably induced by high-dose isoflurane. They conjectured that high and low-frequency neural activities may independently or differentially influence the BOLD signal. Our study also examined the influence of various LFP frequency bands on the BOLD signal and found inversed LFP-BOLD relationship between low- and high-frequency LFP powers. We also added more results on the analysis of infraslow LFP signals. Regardless, since the reference study did not examine the spatial relationship of LFP and BOLD activities, we cannot comment on how it may provide insight into our results.

https://pubmed.ncbi.nlm.nih.gov/26041826/ This paper found electrophysiological correlates within the BOLD signal when using BOLD analysis methods not used in the reviewed paper, and furthermore that some of these correlate with electrophysiological frequencies not studied in the reviewed paper (< 1 Hz).

We have added more results on the analysis of infraslow LFP signals and acknowledged the value of dynamic rsfMRI analysis in studies of BOLDelectrophysiology relationship.

I am not saying the authors need to use all these methods or even cite these papers. As I stated in their review, they merely need to (1) cite some of the most relevant for the proper context, the above list can maybe help (2) remove the claim of an "electrophysiology invisible signal" (3) use terms more commonly used in these papers for the extent of correlation with the electrode, other than "spatial variance."

We thank the reviewer again for providing a detailed list of reference studies. We have added the related discussion to the revised manuscript as described above.

Comment 2: The abstract entirely and much of the rest of the paper should be rewritten to be more reasonable. The authors would do well to review some of the past controversies in this area, e.g. Magri et al. J Neurosci. 2012 Jan 25;32(4):1395-407.

We have made significant revision to improve the writing of the paper. The reference paper has been added to the revised manuscript.

Comment 3: This should be re-written and the terminology used here should be chosen more carefully.

The writing of the manuscript has been improved with more careful choice of terminology.

Major method problem:Comment 4: At a minimum, the authors should be transforming the uniform distribution of CC results to Z or T values and using randn() instead of rand() in MATLAB.

Below is the figure illustrating the simulation results by transforming CC values to Z score. Results obtained remain consistent.

**Author response image 2. sa3fig2:** 

Minor problems:Comment 5: "MR-510 compatible electrodes (MRCM16LP, NeuroNexus Inc)"Details of this type of electrode are not readily available. But for studies like this one, further information on materials is critical as this determines the frequency coverage, which is not even across all LFP frequencies for all materials. Most commercially prepared electrodes cannot record <1Hz accurately, and this study includes at least 0.11Hz in some of its analysis.

The type of electrode used in our current study is a silicon-based micromachined probe. These probes are fabricated using photolithographic techniques to pattern thin layers of conductive materials onto a silicon substrate. This probe is capable of recording the LFP activity within a broad frequency range, starting from 0.1Hz . We added this information to the revised manuscript.

Comment 6: Grounding to the cerebellum in theory would remove global conduction from the LFP but also global signal regression is done to the fMRI. Does the LFP-rsfMRI correlation change due to the regression or does only the rsfMRI-rsfMRI correlation change?

The results obtained with global signal regression were consistent with those obtained without it (see Figs. S4-S5), and therefore, we do not believe our results are affected by this preprocessing step.

Comment 7. Avoid colloquial language like "on the other hand" etc.

We used more appropriate language in the revised manuscript.

References:

Bolt, T., Nomi, J.S., Bzdok, D., Salas, J.A., Chang, C., Thomas Yeo, B.T., Uddin, L.Q., Keilholz, S.D., 2022. A parsimonious description of global functional brain organization in three spatiotemporal patterns. Nat Neurosci 25, 1093-1103.

Cabral, J., Fernandes, F.F., Shemesh, N., 2023. Intrinsic macroscale oscillatory modes driving long range functional connectivity in female rat brains detected by ultrafast fMRI. Nat Commun 14, 375.

Hacker, C.D., Snyder, A.Z., Pahwa, M., Corbetta, M., Leuthardt, E.C., 2017. Frequencyspecific electrophysiologic correlates of resting state fMRI networks. Neuroimage 149, 446-457.

Kucyi, A., Schrouff, J., Bickel, S., Foster, B.L., Shine, J.M., Parvizi, J., 2018. Intracranial Electrophysiology Reveals Reproducible Intrinsic Functional Connectivity within Human Brain Networks. J Neurosci 38, 4230-4242.

Li, J.M., Acland, B.T., Brenner, A.S., Bentley, W.J., Snyder, L.H., 2022. Relationships between correlated spikes, oxygen and LFP in the resting-state primate. Neuroimage 247, 118728.

Ma, Y., Shaik, M.A., Kozberg, M.G., Kim, S.H., Portes, J.P., Timerman, D., Hillman, E.M., 2016. Resting-state hemodynamics are spatiotemporally coupled to synchronized and symmetric neural activity in excitatory neurons. Proc Natl Acad Sci U S A 113, E8463-E8471.

Ma, Z., Zhang, N., 2018. Temporal transitions of spontaneous brain activity. Elife 7.

Shi, Z., Wu, R., Yang, P.F., Wang, F., Wu, T.L., Mishra, A., Chen, L.M., Gore, J.C., 2017. High spatial correspondence at a columnar level between activation and resting state fMRI signals and local field potentials. Proc Natl Acad Sci U S A 114, 52535258.

Thompson, G.J., Pan, W.J., Magnuson, M.E., Jaeger, D., Keilholz, S.D., 2014. Quasiperiodic patterns (QPP): large-scale dynamics in resting state fMRI that correlate with local infraslow electrical activity. Neuroimage 84, 1018-1031.

Uhlirova, H., Kilic, K., Tian, P., Thunemann, M., Desjardins, M., Saisan, P.A., Sakadzic, S., Ness, T.V., Mateo, C., Cheng, Q., Weldy, K.L., Razoux, F., Vandenberghe, M.,

Cremonesi, J.A., Ferri, C.G., Nizar, K., Sridhar, V.B., Steed, T.C., Abashin, M.,

Fainman, Y., Masliah, E., Djurovic, S., Andreassen, O.A., Silva, G.A., Boas, D.A., Kleinfeld, D., Buxton, R.B., Einevoll, G.T., Dale, A.M., Devor, A., 2016. Cell type specificity of neurovascular coupling in cerebral cortex. Elife 5.

Vafaii, H., Mandino, F., Desrosiers-Gregoire, G., O'Connor, D., Markicevic, M., Shen, X.,

Ge, X., Herman, P., Hyder, F., Papademetris, X., Chakravarty, M., Crair, M.C., Constable, R.T., Lake, E.M.R., Pessoa, L., 2024. Multimodal measures of spontaneous brain activity reveal both common and divergent patterns of cortical functional organization. Nat Commun 15, 229.